# Estimating the Rate-Distortion Function by Wasserstein Gradient Descent

**Yibo Yang**[1]    **Stephan Eckstein**[2]    **Marcel Nutz**[3]    **Stephan Mandt**[1]
[1]University of California, Irvine    [2]ETH Zurich    [3]Columbia University
{yibo.yang, mandt}@uci.edu
stephan.eckstein@math.ethz.ch
mnutz@columbia.edu

## Abstract

In the theory of lossy compression, the rate-distortion (R-D) function $R(D)$ describes how much a data source can be compressed (in bit-rate) at any given level of fidelity (distortion). Obtaining $R(D)$ for a given data source establishes the fundamental performance limit for all compression algorithms. We propose a new method to estimate $R(D)$ from the perspective of optimal transport. Unlike the classic Blahut–Arimoto algorithm which fixes the support of the reproduction distribution in advance, our Wasserstein gradient descent algorithm learns the support of the optimal reproduction distribution by moving particles. We prove its local convergence and analyze the sample complexity of our R-D estimator based on a connection to entropic optimal transport. Experimentally, we obtain comparable or tighter bounds than state-of-the-art neural network methods on low-rate sources while requiring considerably less tuning and computation effort. We also highlight a connection to maximum-likelihood deconvolution and introduce a new class of sources that can be used as test cases with known solutions to the R-D problem.

## 1   Introduction

The rate-distortion (R-D) function $R(D)$ occupies a central place in the theory of lossy compression. For a given data source and a fidelity (or *distortion*) criterion, $R(D)$ characterizes the minimum possible communication cost needed to reproduce a source sample within an error threshold of $D$, by *any* compression algorithm [Shannon, 1959]. A basic scientific and practical question is therefore establishing $R(D)$ for any given data source of interest, which helps assess the (sub)optimality of the compression algorithms and guide their development. The classic algorithm by Blahut [1972] and Arimoto [1972] assumes a known discrete source and computes its $R(D)$ by an exhaustive optimization procedure. This often has limited applicability in practice, and a line of research has sought to instead *estimate* $R(D)$ from data samples [Harrison and Kontoyiannis, 2008, Gibson, 2017], with recent methods [Yang and Mandt, 2022, Lei et al., 2023a] inspired by deep generative models.

In this work, we propose a new approach to R-D estimation from the perspective of optimal transport. Our starting point is the formulation of the R-D problem as the minimization of a certain rate functional [Harrison and Kontoyiannis, 2008] over the space of probability measures on the reproduction alphabet. Optimization over such an infinite-dimensional space has long been studied under gradient flows [Ambrosio et al., 2008], and we consider a concrete algorithmic implementation based on moving particles in space. This formulation of the R-D problem also suggests connections to entropic optimal transport and non-parametric statistics, each offering us new insight into the solution of the R-D problem under a quadratic distortion. More specifically, our contributions are three-fold:

First, we introduce a neural-network-free $R(D)$ upper bound estimator for continuous alphabets. We implement the estimator by Wasserstein gradient descent (WGD) over the space of reproduction

distributions. Experimentally, we found the method to converge much more quickly than state-of-the-art neural methods with hand-tuned architectures, while offering comparable or tighter bounds.

Second, we theoretically characterize convergence of our WGD algorithm and the sample complexity of our estimator. The latter draws on a connection between the R-D problem and that of minimizing an entropic optimal transport (EOT) cost relative to the source measure, allowing us to turn statistical bounds for EOT [Mena and Niles-Weed, 2019] into finite-sample bounds for R-D estimation.

Finally, we introduce a new, rich class of sources with known ground truth, including Gaussian mixtures, as a benchmark for algorithms. While the literature relies on the Gaussian or Bernoulli for this purpose, we use the connection with maximum likelihood deconvolution to show that a Gaussian convolution of *any* distribution can serve as a source with a known solution to the R-D problem.

## 2 Lossy compression, entropic optimal transport, and MLE

This section introduces the R-D problem and its rich connections to entropic optimal transport and statistics, along with new insights into its solution. Sec. 2.1 sets the stage for our method (Sec. 4) by a known formulation of the standard R-D problem as an optimization problem over a space of probability measures. Sec. 2.2 discusses the equivalence between the R-D problem and a projection of the source distribution under entropic optimal transport; this is a key to our sample complexity results in Sec. 4.3. Lastly, Sec 2.3 gives a statistical interpretation of R-D as maximum-likelihood deconvolution and uses it to analytically derive a segment of the R-D curve for a new class of sources under quadratic distortion; this allows us to assess the optimality of algorithms in experiment Sec. 5.1.

### 2.1 Setup

For a memoryless data source $X$ with distribution $P_X$, its rate-distortion (R-D) function describes the minimum possible number of bits per sample needed to reproduce the source within a prescribed distortion threshold $D$. Let the source and reproduction take values in two sets $\mathcal{X}$ and $\mathcal{Y}$, known as the source and reproduction *alphabets*, and let $\rho : (\mathcal{X}, \mathcal{Y}) \to [0, \infty)$ be a given distortion function. The R-D function is defined by the following optimization problem [Polyanskiy and Wu, 2022],

$$R(D) = \inf_{Q_{Y|X} : \mathbb{E}_{P_X Q_{Y|X}}[\rho(X,Y)] \leq D} I(X;Y), \qquad (1)$$

where $Q_{Y|X}$ is any Markov kernel from $\mathcal{X}$ to $\mathcal{Y}$ conceptually associated with a (possibly) stochastic compression algorithm, and $I(X;Y)$ is the mutual information of the joint distribution $P_X Q_{Y|X}$.

For ease of presentation, we now switch to a more abstract notation without reference to random variables. We provide the precise definitions in the Supplementary Material. Let $\mathcal{X}$ and $\mathcal{Y}$ be standard Borel spaces; let $\mu \in \mathcal{P}(\mathcal{X})$ be a fixed probability measure on $\mathcal{X}$, which should be thought of as the source distribution $P_X$. For a measure $\pi$ on the product space $\mathcal{X} \times \mathcal{Y}$, the notation $\pi_1$ (or $\pi_2$) denotes the first (or second) marginal of $\pi$. For any $\nu \in \mathcal{P}(\mathcal{Y})$, we denote by $\Pi(\mu, \nu)$ the set of couplings between $\mu$ and $\nu$ (i.e., $\pi_1 = \mu$ and $\pi_2 = \nu$). Similarly, $\Pi(\mu, \cdot)$ denotes the set of measures $\pi$ with $\pi_1 = \mu$. Throughout the paper, $K$ denotes a transition kernel (conditional distribution) from $\mathcal{X}$ to $\mathcal{Y}$, and $\mu \otimes K$ denotes the product measure formed by $\mu$ and $K$. Then $R(D)$ is equivalent to

$$R(D) = \inf_{K : \int \rho d(\mu \otimes K) \leq D} H(\mu \otimes K | \mu \otimes (\mu \otimes K)_2) = \inf_{\pi \in \Pi(\mu, \cdot) : \int \rho d\pi \leq D} H(\pi | \pi_1 \otimes \pi_2), \quad (2)$$

where $H$ denotes relative entropy, i.e., for two measures $\alpha, \beta$ defined on a common measurable space, $H(\alpha|\beta) := \int \log(\frac{d\alpha}{d\beta}) d\alpha$ when $\alpha$ is absolutely continuous w.r.t $\beta$, and infinite otherwise.

To make the problem more tractable, we follow the approach of the classic Blahut–Arimoto algorithm [Blahut, 1972, Arimoto, 1972] (to be discussed in Sec. 3.1) and work with an equivalent unconstrained Lagrangian problem as follows. Instead of parameterizing the R-D function via a distortion threshold $D$, we parameterize it via a Lagrange multiplier $\lambda \geq 0$. For each fixed $\lambda$ (usually selected from a predefined grid), we aim to solve the following optimization problem,

$$F_\lambda(\mu) := \inf_{\nu \in \mathcal{P}(\mathcal{Y})} \inf_{\pi \in \Pi(\mu, \cdot)} \lambda \int \rho d\pi + H(\pi | \mu \otimes \nu). \qquad (3)$$

Geometrically, $F_\lambda(\mu) \in \mathbb{R}$ is the y-axis intercept of a tangent line to the $R(D)$ with slope $-\lambda$, and $R(D)$ is determined by the convex envelope of all such tangent lines [Gray, 2011]. To simplify notation, we often drop the dependence on $\lambda$ (e.g., we write $F(\mu) = F_\lambda(\mu)$) whenever it is harmless.

To set the stage for our later developments, we write the unconstrained R-D problem as

$$F_\lambda(\mu) = \inf_{\nu \in \mathcal{P}(\mathcal{Y})} \mathcal{L}_{BA}(\mu, \nu), \tag{4}$$

$$\mathcal{L}_{BA}(\mu, \nu) := \inf_{\pi \in \Pi(\mu, \cdot)} \lambda \int \rho d\pi + H(\pi | \mu \otimes \nu) = \inf_K \lambda \int \rho d(\mu \otimes K) + H(\mu \otimes K | \mu \otimes \nu), \tag{5}$$

where we refer to the optimization objective $\mathcal{L}_{BA}$ as the *rate function* [Harrison and Kontoyiannis, 2008]. We abuse the notation to write $\mathcal{L}_{BA}(\nu) := \mathcal{L}_{BA}(\mu, \nu)$ when it is viewed as a function of $\nu$ only, and refer to it as the *rate functional*. The rate function characterizes a generalized Asymptotic Equipartition Property, where $\mathcal{L}_{BA}(\mu, \nu)$ is the asymptotically optimal cost of lossy compression of data $X \sim \mu$ using a random codebook constructed from samples of $\nu$ [Dembo and Kontoyiannis, 2002]. Notably, the optimization in (5) can be solved analytically [Csiszár, 1974a, Lemma 1.3], and $\mathcal{L}_{BA}$ simplifies to

$$\mathcal{L}_{BA}(\mu, \nu) = \int_{\mathcal{X}} -\log\left(\int_{\mathcal{Y}} e^{-\lambda \rho(x,y)} \nu(dy)\right) \mu(dx). \tag{6}$$

In practice, the source $\mu$ is only accessible via independent samples, on the basis of which we propose to estimate its $R(D)$, or equivalently $F(\mu)$. Let $\mu^m$ denote an $m$-sample empirical measure of $\mu$, i.e., $\mu^m = \sum_{i=1}^m \delta_{x_i}$ with $x_{1,\dots,n}$ being independent samples from $\mu$, which should be thought of as the "training data". Following Harrison and Kontoyiannis [2008], we consider two kinds of (plug-in) estimators for $F(\mu)$: (1) the non-parametric estimator $F(\mu^m)$, and (2) the parametric estimator $F^{\mathcal{H}}(\mu^m) := \inf_{\nu \in \mathcal{H}} \mathcal{L}_{BA}(\mu^m, \nu)$, where $\mathcal{H}$ is a family of probability measures on $\mathcal{Y}$. Harrison and Kontoyiannis [2008] showed that under rather broad conditions, both kinds of estimators are strongly consistent, i.e., $F(\mu^m)$ converges to $F(\mu)$ (and respectively, $F^{\mathcal{H}}(\mu^m)$ to $F^{\mathcal{H}}(\mu)$) with probability one as $m \to \infty$. Our algorithm will implement the parametric estimator $F^{\mathcal{H}}(\mu^m)$ with $\mathcal{H}$ chosen to be the set of probability measures with finite support, and we will develop finite-sample convergence results for both kinds of estimators in the continuous setting (Proposition. 4.3).

## 2.2 Connection to entropic optimal transport

The R-D problem turns out to have a close connection to entropic optimal transport (EOT) [Peyré and Cuturi, 2019], which we will exploit in Sec. 4.3 to obtain sample complexity results under our approach. For $\epsilon > 0$, the entropy-regularized optimal transport problem is given by

$$\mathcal{L}_{EOT}(\mu, \nu) := \inf_{\pi \in \Pi(\mu, \nu)} \int \rho d\pi + \epsilon H(\pi | \mu \otimes \nu). \tag{7}$$

We now consider the problem of projecting $\mu$ onto $\mathcal{P}(\mathcal{Y})$ under the cost $\mathcal{L}_{EOT}$:

$$\inf_{\nu \in \mathcal{P}(\mathcal{Y})} \mathcal{L}_{EOT}(\mu, \nu). \tag{8}$$

In the OT literature this is known as the (regularized) Kantorovich estimator [Bassetti et al., 2006] for $\mu$, and can also be viewed as a Wasserstein barycenter problem [Agueh and Carlier, 2011].

With the identification $\epsilon = \lambda^{-1}$, problem (8) is in fact equivalent to the R-D problem (4): compared to $\mathcal{L}_{BA}$ (5), the extra constraint on the second marginal of $\pi$ in $\mathcal{L}_{EOT}$ (7) is redundant at the optimal $\nu$. More precisely, Lemma 7.1 shows that (we omit the notational dependence on $\mu$ when it is fixed):

$$\inf_{\nu \in \mathcal{P}(\mathcal{Y})} \mathcal{L}_{EOT}(\nu) = \inf_{\nu \in \mathcal{P}(\mathcal{Y})} \lambda^{-1} \mathcal{L}_{BA}(\nu) \qquad \text{and} \qquad \arg\min_{\nu \in \mathcal{P}(\mathcal{Y})} \mathcal{L}_{EOT}(\nu) = \arg\min_{\nu \in \mathcal{P}(\mathcal{Y})} \mathcal{L}_{BA}(\nu). \tag{9}$$

Existence of a minimizer holds under mild conditions, for instance if $\mathcal{X} = \mathcal{Y} = \mathbb{R}^d$ and $\rho(x, y)$ is a coercive lower semicontinuous function of $y - x$ [Csiszár, 1974a, p. 66].

## 2.3 Connection to maximum-likelihood deconvolution

The connection between R-D and maximum-likelihood estimation has been observed in the information theory, machine learning and compression literature [Harrison and Kontoyiannis, 2008, Alemi et al., 2018, Ballé et al., 2017, Theis et al., 2017, Yang et al., 2020, Yang and Mandt, 2022]. Here, we bring attention to a basic equivalence between the R-D problem and maximum-likelihood deconvolution, where the connection is particularly natural under a quadratic distortion function. Also see [Rigollet and Weed, 2018] for a related discussion that inspired ours and extension to a non-quadratic distortion. We provide further insight from the view of variational learning and inference in Section 10.

Maximum-likelihood deconvolution is a classical problem of non-parametric statistics and mixture models [Carroll and Hall, 1988, Lindsay and Roeder, 1993]. The deconvolution problem is concerned with estimating an unknown distribution $\alpha$ from noise-corrupted observations $X_1, X_2, ...$, where for each $i \in \mathbb{N}$, we have $X_i = Y_i + N_i$, $Y_i \overset{i.i.d.}{\sim} \alpha$, and $N_i$ are i.i.d. independent noise variables with a known distribution. For concreteness, suppose all variables are $\mathbb{R}^d$ valued and the noise distribution is $\mathcal{N}(0, \sigma^2 I_d)$ with Lebesgue density $\phi_{\sigma^2}$. Denote the distribution of the observations $X_i$ by $\mu$. Then $\mu$ has a Lebesgue density given by the convolution $\alpha * \phi_{\sigma^2}(x) := \int \phi_{\sigma^2}(x - y)\alpha(dy)$. Here, we consider the population-level (instead of the usual sample-based) maximum-likelihood estimator (MLE) for $\alpha$:

$$\nu^* = \arg\max_{\nu \in \mathcal{P}(\mathbb{R}^d)} \int \log\left(\nu * \phi_{\sigma^2}(x)\right) \mu(dx), \quad (10)$$

and observe that $\nu^* = \alpha$. Plugging in the density $\phi_{\sigma^2}(x) \propto e^{-\frac{1}{2\sigma^2}\|x\|^2}$, we see that the MLE problem (10) is equivalent to the R-D problem (4) with $\rho(x, y) = \frac{1}{2}\|x - y\|^2$, $\lambda = \frac{1}{\sigma^2}$, and $\mathcal{L}_{BA}$ given by (6) in the form of a marginal log-likelihood. Thus the R-D problem has the interpretation of estimating a distribution from its noisy observations given through $\mu$, assuming a Gaussian noise with variance $\frac{1}{\lambda}$.

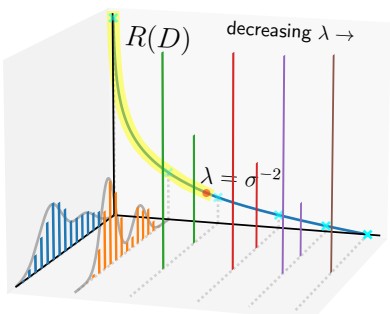

Figure 1: The $R(D)$ of a Gaussian mixture source, and the estimated optimal reproduction distributions $\nu^*$ (in bar plots) at varying R-D trade-offs. For any $\lambda \in [\sigma^{-2}, \infty)$, the corresponding $R(D)$ (yellow segment) is known analytically as is the optimal reproduction distribution $\nu^*$ (whose density is plotted in gray). For $\lambda \in (0, \sigma^{-2}]$, $\nu^*$ becomes singular and concentrated on two points, collapsing to the source mean as $\lambda \to 0$.

This connection suggests analytical solutions to the R-D problem for a variety of sources that arise from convolving an underlying distribution with Gaussian noise. Consider an R-D problem (4) with $\mathcal{X} = \mathcal{Y} = \mathbb{R}^d$, $\rho(x, y) = \frac{1}{2}\|x - y\|^2$, and let the source $\mu$ be the convolution between an arbitrary measure $\alpha \in \mathcal{P}(\mathcal{Y})$ and Gaussian noise with known variance $\sigma^2$. E.g., using a discrete measure for $\alpha$ results in a Gaussian mixture source with equal covariance among its components. When $\lambda = \frac{1}{\sigma^2}$, we recover exactly the population-MLE problem (10) discussed earlier, which has the solution $\nu^* = \alpha$. While this allows us to obtain one point of $R(D)$, we can in fact extend this idea to any $\lambda \geq \frac{1}{\sigma^2}$ and obtain the analytical form for the corresponding *segment* of the R-D curve. Specifically, for any $\lambda \geq \frac{1}{\sigma^2}$, applying the summation rule for independent Gaussians reveals the source distribution $\mu$ as

$$\mu = \alpha * \mathcal{N}(0, \sigma^2) = \alpha * \mathcal{N}(0, \sigma^2 - \frac{1}{\lambda}) * \mathcal{N}(0, \frac{1}{\lambda}) = \alpha_\lambda * \mathcal{N}(0, \frac{1}{\lambda}), \quad \alpha_\lambda := \alpha * \mathcal{N}(0, \sigma^2 - \frac{1}{\lambda}),$$

i.e., as the convolution between another underlying distribution $\alpha_\lambda$ and independent noise with variance $\frac{1}{\lambda}$. A solution to the R-D problem (3) is then analogously given by $\nu^* = \alpha_\lambda$, with the corresponding optimal coupling given by $\nu^* \otimes \tilde{K}$, $\tilde{K}(y, dx) = \mathcal{N}(y, \frac{1}{\lambda})$. [1] Evaluating the distortion and mutual information of the coupling then yields the $R(D)$ point associated with $\lambda$. Fig. 1 illustrates the $R(D)$ of a toy Gaussian mixture source, along with the $\nu^*$ estimated by our proposed WGD algorithm (Sec. 4); note that $\nu^*$ transitions from continuous (a Gaussian mixture with smaller component variances) to singular (a mixture of two Diracs) at $\lambda = \sigma^{-2}$. See caption for more details.

## 3 Related Work

### 3.1 Blahut–Arimoto

The Blahut–Arimoto (BA) algorithm [Blahut, 1972, Arimoto, 1972] is the default method for computing $R(D)$ for a known and discrete case. For a fixed $\lambda$, BA carries out the optimization problem (3) via coordinate ascent. Starting from an initial measure $\nu^{(0)} \in \mathcal{P}(\mathcal{Y})$, the BA algorithm at step $t$ computes an updated pair $(\nu^{(t+1)}, K^{(t+1)})$ as follows

$$\frac{dK^{(t+1)}(x, \cdot)}{d\nu^{(t)}}(y) = \frac{e^{-\lambda\rho(x,y)}}{\int e^{-\lambda\rho(x,y')}\nu^{(t)}(dy')}, \quad \forall x \in \mathcal{X}, \quad (11)$$

$$\nu^{(t+1)} = (\mu \otimes K^{(t+1)})_2. \quad (12)$$

---

[1] Here $\tilde{K}$ maps from the reproduction to the source alphabet, opposite to the kernel $K$ elsewhere in the text.

When the alphabets are finite, the above computation can be carried out in matrix and vector operations, and the resulting sequence $\{(\nu^{(t)}, K^{(t)})\}_{t=1}^{\infty}$ can be shown to converge to an optimum of (3); cf. [Csiszár, 1974b, Csiszár, 1984]. When the alphabets are not finite, e.g., $\mathcal{X} = \mathcal{Y} = \mathbb{R}^d$, the BA algorithm no longer applies, as it is unclear how to digitally represent the measure $\nu$ and kernel $K$ and to tractably perform the integrals required by the algorithm. The common workaround is to perform a discretization step and then apply BA on the resulting discrete problem.

One standard discretization method is to tile up the alphabets with small bins [Gray and Neuhoff, 1998]. This quickly becomes infeasible as the number of dimensions increases. We therefore consider discretizing the data space $\mathcal{X}$ to be the support of training data distribution $\mu^m$, i.e., the discretized alphabet is the set of training samples; this can be justified by the consistency of the parametric R-D estimator $F^{\mathcal{H}}(\mu^m)$ [Harrison and Kontoyiannis, 2008]. It is less clear how to discretize the reproduction space $\mathcal{Y}$, especially in high dimensions. Since we work with $\mathcal{X} = \mathcal{Y}$, we will disretize $\mathcal{Y}$ similarly and use an $n$-element random subset of the training samples, as also considered by Lei et al. [2023a]. As we will show, this rather arbitrary placement of the support of $\nu$ results in poor performance, and can be significantly improved from our perspective of evolving particles.

### 3.2 Neural network-based methods for estimating $R(D)$

**RD-VAE ([Yang and Mandt, 2022]):** To overcome the limitations of the BA algorithm, Yang and Mandt [2022] proposed to parameterize the transition kernel $K$ and reproduction distribution $\nu$ of the BA algorithm by neural density estimators [Papamakarios et al., 2021], and optimize the same objective (3) by (stochastic) gradient descent. They estimate (3) by Monte Carlo using joint samples $(X_i, Y_i) \sim \mu \otimes K$; in particular, the relative entropy can be written as $H(\mu \otimes K | \mu \otimes \nu) = \int \int \log \left( \frac{dK(x,\cdot)}{d\nu}(y) \right) K(x, dy) \mu(dx)$, where the integrand is computed exactly via a density ratio. In practice, an alternative parameterization is often used where the neural density estimators are defined on a lower dimensional latent space than the reproduction alphabet, and the resulting approach is closely related to VAEs [Kingma and Welling, 2013]. Yang and Mandt [2022] additionally propose a neural estimator for a lower bound on $R(D)$, based on a dual representation due to Csiszár [1974a].

**NERD [Lei et al., 2023a]:** Instead of working with the transition kernel $K$ as in the RD-VAE, Lei et al. [2023a] considered optimizing the form of the rate functional in (6), via gradient descent on the parameters of $\nu$ parameterized by a neural network. Let $\nu^{\mathcal{Z}}$ be a base distribution over $\mathcal{Z} = \mathbb{R}^K$, such as the standard Gaussian, and $\omega : \mathcal{Z} \to \mathcal{Y}$ be a decoder network. The variational measure $\nu$ is then modeled as the image measure of $\nu^{\mathcal{Z}}$ under $\omega$. To evaluate and optimize the objective (6), the intractable inner integral w.r.t. $\nu$ is replaced with a plug-in estimator, so that for a given $x \in \mathcal{X}$,

$$-\log \left( \int_{\mathcal{Y}} e^{-\lambda \rho(x,y)} \nu(dy) \right) \approx -\log \left( \frac{1}{n} \sum_{j=1}^{n} e^{-\lambda \rho(x,Y_j)} \right), \quad Y_j \sim \nu, j = 1, 2, ..., n. \quad (13)$$

After training, we estimate an R-D upper bound using $n$ samples from $\nu$ (to be discussed in Sec. 4.4).

### 3.3 Other related work

**Within information theory:** Recent work by Wu et al. [2022] and Lei et al. [2023b] also note the connection between the R-D function and entropic optimal transport. Wu et al. [2022] compute the R-D function in the finite and known alphabet setting by solving a version of the EOT problem (8), whereas Lei et al. [2023b] numerically verify the equivalence (9) on a discrete problem and discuss the connection to scalar quantization. We also experimented with estimating $R(D)$ by solving the EOT problem (8), but found it computationally much more efficient to work with the rate functional (6), and we see the primary benefit of the EOT connection as bringing in tools from statistical OT [Genevay et al., 2019, Mena and Niles-Weed, 2019, Rigollet and Stromme, 2022] for R-D estimation. **Outside of information theory:** Rigollet and Weed [2018] note a connection between the EOT projection problem (8) and maximum-likelihood deconvolution (10); our work complements their perspective by re-interpreting both problems through the equivalent R-D problem. Unbeknownst to us at the time, Yan et al. [2023] proposed similar algorithms to ours in the context of Gaussian mixture estimation, which we recognize as R-D estimation under quadratic distortion (see Sec. 2.3). Their work is based on gradient flow in the Fisher-Rao-Wasserstein (FRW) geometry [Chizat et al., 2018], which our hybrid algorithm can be seen as implementing. Yan et al. [2023] prove that, in an

idealized setting with infinite particles, FRW gradient descent does not get stuck at local minima; by contrast, our convergence and sample-complexity results (Prop. 4.2, 4.3) hold for any finite number of particles. We additionally consider larger-scale problems and the stochastic optimization setting.

# 4 Proposed method

For our algorithm, we require $\mathcal{X} = \mathcal{Y} = \mathbb{R}^d$ and $\rho$ be continuously differentiable. We now introduce the gradient descent algorithm in Wasserstein space to solve the problems (4) and (8). We defer all proofs to the Supplementary Material. To minimize a functional $\mathcal{L} : \mathcal{P}(\mathcal{Y}) \to \mathbb{R}$ over the space of probability measures, our algorithm essentially simulates the gradient flow [Ambrosio et al., 2008] of $\mathcal{L}$ and follows the trajectory of steepest descent in the Wasserstein geometry. In practice, we represent a measure $\nu^{(t)} \in \mathcal{P}(\mathcal{Y})$ by a collection of particles and at each time step update $\nu^{(t)}$ in a direction of steepest descent of $\mathcal{L}$ as given by its (negative) *Wasserstein gradient*. Denote by $\mathcal{P}_n(\mathbb{R}^d)$ the set of probability measures on $\mathbb{R}^d$ that are supported on at most $n$ points. Our algorithm implements the parametric R-D estimator with the choice $\mathcal{H} = \mathcal{P}_n(\mathbb{R}^d)$ (see discussions at the end of Sec. 2.1).

## 4.1 Wasserstein gradient descent (WGD)

Abstractly, Wasserstein gradient descent updates the variational measure $\nu$ to its pushforward $\tilde{\nu}$ under the map $(\mathrm{id} - \gamma \Psi)$, for a function $\Psi : \mathbb{R}^d \to \mathbb{R}^d$ called the Wasserstein gradient of $\mathcal{L}$ at $\nu$ (see below) and a step size $\gamma$. To implement this scheme, we represent $\nu$ as a convex combination of Dirac measures, $\nu = \sum_{i=1}^{n} w_i \delta_{x_i}$ with locations $\{x_i\}_{i=1}^{n} \subset \mathbb{R}^d$ and weights $\{w_i\}_{i=1}^{n}$. The algorithm moves each particle $x_i$ in the direction of $-\Psi(x_i)$, more precisely, $\tilde{\nu} = \sum_{i=1}^{n} w_i \delta_{x_i - \gamma \Psi(x_i)}$.

---

**Algorithm 1** Wasserstein gradient descent

**Inputs:** Loss function $\mathcal{L} \in \{\mathcal{L}_{BA}, \mathcal{L}_{EOT}\}$; data distribution $\mu \in \mathcal{P}(\mathbb{R}^d)$; the number of particles $n \in \mathbb{N}$; total number of iterations $N \in \mathbb{N}$; step sizes $\gamma_1, \ldots, \gamma_N$; batch size $m \in \mathbb{N}$.
**for** $t = 1, \ldots, N$ **do**
    Pick an initial measure $\nu^{(0)} \in \mathcal{P}_n(\mathbb{R}^d)$, e.g., setting the particles to $n$ random samples from $\mu$.
    **if** support of $\mu$ contains more than $m$ points **then**
        $\mu^m \leftarrow \frac{1}{m} \sum_{i=1}^{m} \delta_{x_i}$ for $x_1, \ldots, x_m$ independent samples from $\mu$
        $\Psi^{(t)} \leftarrow$ Wasserstein gradient of $\mathcal{L}(\mu^m, \cdot)$ at $\nu^{(t-1)}$ {see Definition 4.1}
    **else**
        $\Psi^{(t)} \leftarrow$ Wasserstein gradient of $\mathcal{L}(\mu, \cdot)$ at $\nu^{(t-1)}$ {see Definition 4.1}
    **end if**
    $\nu^{(t)} \leftarrow \left(\mathrm{id} - \gamma_t \Psi^{(t)}\right)_{\#} \nu^{(t-1)}$ {"#" denotes pushforward}
**end for**
**Return:** $\nu^{(N)}$

---

Since the optimization objectives (4) and (8) appear as integrals w.r.t. the data distribution $\mu$, we can also apply stochastic optimization and perform stochastic gradient descent on mini-batches with size $m$. This allows us to handle a very large or infinite amount of data samples, or when the source is continuous. We formalize the procedure in Algorithm 1.

The following gives a constructive definition of a Wasserstein gradient which forms the computational basis of our algorithm. In the literature, the Wasserstein gradient is instead usually defined as a Fréchet differential (cf. [Ambrosio et al., 2008, Definition 10.1.1]), but we emphasize that in smooth settings, the given definition recovers the one from the literature (cf. [Chizat, 2022, Lemma A.2]).

**Definition 4.1.** *For a functional $\mathcal{L} : \mathcal{P}(\mathcal{Y}) \to \mathbb{R}$ and $\nu \in \mathcal{P}(\mathcal{Y})$, we say that $V_{\mathcal{L}}(\nu) : \mathbb{R}^d \to \mathbb{R}$ is a first variation of $\mathcal{L}$ at $\nu$ if*

$$\lim_{\varepsilon \to 0} \frac{\mathcal{L}((1-\varepsilon)\nu + \varepsilon \tilde{\nu}) - \mathcal{L}(\nu)}{\varepsilon} = \int V_{\mathcal{L}}(\nu) \, d(\tilde{\nu} - \nu) \quad \text{for all } \tilde{\nu} \in \mathcal{P}(\mathcal{Y}).$$

*We call its (Euclidean) gradient $\nabla V_{\mathcal{L}}(\nu) : \mathbb{R}^d \to \mathbb{R}^d$, if it exists, the Wasserstein gradient of $\mathcal{L}$ at $\nu$.*

For $\mathcal{L} = \mathcal{L}_{EOT}$, the first variation is given by the Kantorovich potential, which is the solution of the convex dual of $\mathcal{L}_{EOT}$ and commonly computed by Sinkhorn's algorithm [Peyré and Cuturi, 2019,

Nutz, 2021]. Specifically, let $(\varphi^\nu, \psi^\nu)$ be potentials for $\mathcal{L}_{EOT}(\mu, \nu)$. Then $V_{\mathcal{L}}(\nu) = \psi^\nu$ is the first variation w.r.t. $\nu$ (cf. [Carlier et al., 2022, equation (20)]), and hence $\nabla \psi^\nu$ is the Wasserstein gradient. This gradient exists whenever $\rho$ is differentiable and the marginals are sufficiently light-tailed; we give details in Sec. 9.1 of the Supplementary Material. For $\mathcal{L} = \mathcal{L}_{BA}$, the first variation can be computed explicitly. As derived in Sec. 9.1 of the Supplementary Material, the first variation at $\nu$ is

$$\psi^\nu(y) = \int -\frac{\exp(-\lambda\rho(x,y))}{\int \exp(-\lambda\rho(x,\tilde{y}))\nu(d\tilde{y})}\mu(dx)$$

and then the Wasserstein gradient is $\nabla \mathcal{L}_{BA}(\nu) = \nabla \psi^\nu$. We observe that $\psi^\nu(y)$ is computationally cheap; it corresponds to running a single iteration of Sinkhorn's algorithm. By contract, finding the potential for $\mathcal{L}_{EOT}$ requires running Sinkhorn's algorithm to convergence.

Like the usual Euclidean gradient, the Wasserstein gradient can be shown to possess a linearization property, whereby the loss functional is reduced by taking a small enough step along its Wasserstein gradient. Following [Carlier et al., 2022], we state it as follows: for any $\tilde{\nu} \in \mathcal{P}(\mathcal{Y})$ and $\pi \in \Pi(\nu, \tilde{\nu})$,

$$\mathcal{L}(\tilde{\nu}) - \mathcal{L}(\nu) = \int (y-x)^\top \nabla V_{\mathcal{L}}(\nu)(x)\,\pi(dx,dy) + o\left(\int \|y-x\|^2\,\pi(dx,dy)\right),$$

$$\left|\int \|\nabla V_{\mathcal{L}}(\nu)\|^2\,d\nu - \int \|\nabla V_{\mathcal{L}}(\tilde{\nu})\|^2\,d\tilde{\nu}\right| \le C W_2(\nu,\tilde{\nu}). \tag{14}$$

The first line of (14) is proved for $\mathcal{L}_{EOT}$ in [Carlier et al., 2022, Proposition 4.2] in the case that the marginals are compactly supported and $\rho$ is twice continuously differentiable. In this setting, the second line of (14) follows using $a^2 - b^2 = (a+b)(a-b)$ and a combination of boundedness and Lipschitz continuity of $\nabla V_{\mathcal{L}}$, see [Carlier et al., 2022, Proposition 2.2 and Corollary 2.4].

The linearization property given by (14) enables us to show that Wasserstein gradient descent for $\mathcal{L}_{EOT}$ and $\mathcal{L}_{BA}$ converges to a stationary point under mild conditions:

**Proposition 4.2** (Convergence of Wasserstein gradient descent). *Let $\gamma_1 \ge \gamma_2 \ge \cdots \ge 0$ satisfy $\sum_{k=1}^\infty \gamma_k = \infty$ and $\sum_{k=1}^\infty \gamma_k^2 < \infty$. Let $\mathcal{L} : \mathcal{P}(\mathbb{R}^d) \to \mathbb{R}$ be Wasserstein differentiable in the sense that (14) holds. Denoting by $\nu^{(t)}$ the steps in Algorithm 1, and suppose that $\mathcal{L}(\nu^{(0)})$ is finite and $\int \|\nabla V_{\mathcal{L}}(\nu^{(t)})\|^2\,d\nu^{(t)}$ is bounded. Then*

$$\lim_{t\to\infty} \int \|\nabla V_{\mathcal{L}}(\nu^{(t)})\|^2\,d\nu^{(t)} = 0.$$

## 4.2 Hybrid algorithm

A main limitation of the BA algorithm is that the support of $\nu^{(t)}$ is restricted to that of the (possibly bad) initialization $\nu^{(0)}$. On the other hand, Wasserstein gradient descent (Algorithm 1) only evolves the particle locations of $\nu^{(t)}$, but not the weights, which are fixed to be uniform by default. We therefore consider a hybrid algorithm where we alternate between WGD and the BA update steps, allowing us to optimize the particle weights as well. Experimentally, this translates to faster convergence than the base WGD algorithm (Sec. 5.1). Note however, unlike WGD, the hybrid algorithm does not directly lend itself to the stochastic optimization setting, as BA updates on mini-batches no longer guarantee monotonic improvement in the objective and can lead to divergence. We treat the convergence of the hybrid algorithm in the Supplementary Material Sec. 9.4.

## 4.3 Sample complexity

Let $\mathcal{X} = \mathcal{Y} = \mathbb{R}^d$ and $\rho(x,y) = \|x-y\|^2$. Leveraging work on the statistical complexity of EOT [Mena and Niles-Weed, 2019], we obtain finite-sample bounds for the theoretical estimators implemented by WGD in terms of the number of particles and source samples. The bounds hold for both the R-D problem (4) and EOT projection problem (8) as they share the same optimizers (see Sec. 2.2), and strengthen existing asymptotic results for empirical R-D estimators [Harrison and Kontoyiannis, 2008]. We note that a recent result by Rigollet and Stromme [2022] might be useful for deriving alternative bounds under distortion functions other than the quadratic.

**Proposition 4.3.** *Let $\mu$ be $\sigma^2$-subgaussian. Then every optimizer $\nu^*$ of (4) and (8) is also $\sigma^2$-subgaussian. Consider $\mathcal{L} := \mathcal{L}_{EOT}$. For a constant $C_d$ only depending on d, we have*

$$\left| \min_{\nu \in \mathcal{P}(\mathbb{R}^d)} \mathcal{L}(\mu, \nu) - \min_{\nu_n \in \mathcal{P}_n(\mathbb{R}^d)} \mathcal{L}(\mu, \nu_n) \right| \leq C_d \, \epsilon \left( 1 + \frac{\sigma^{\lceil 5d/2 \rceil + 6}}{\epsilon^{\lceil 5d/4 \rceil + 3}} \right) \frac{1}{\sqrt{n}},$$

$$\mathbb{E}\left[ \left| \min_{\nu \in \mathcal{P}(\mathbb{R}^d)} \mathcal{L}(\mu, \nu) - \min_{\nu \in \mathcal{P}(\mathbb{R}^d)} \mathcal{L}(\mu^m, \nu) \right| \right] \leq C_d \, \epsilon \left( 1 + \frac{\sigma^{\lceil 5d/2 \rceil + 6}}{\epsilon^{\lceil 5d/4 \rceil + 3}} \right) \frac{1}{\sqrt{m}},$$

$$\mathbb{E}\left[ \left| \min_{\nu \in \mathcal{P}(\mathbb{R}^d)} \mathcal{L}(\mu, \nu) - \min_{\nu_n \in \mathcal{P}_n(\mathbb{R}^d)} \mathcal{L}(\mu^m, \nu_n) \right| \right] \leq C_d \, \epsilon \left( 1 + \frac{\sigma^{\lceil 5d/2 \rceil + 6}}{\epsilon^{\lceil 5d/4 \rceil + 3}} \right) \left( \frac{1}{\sqrt{m}} + \frac{1}{\sqrt{n}} \right),$$

*for all $n, m \in \mathbb{N}$, where $\mathcal{P}_n(\mathbb{R}^d)$ is the set of probability measures over $\mathbb{R}^d$ supported on at most $n$ points, $\mu^m$ is the empirical measure of $\mu$ with $m$ independent samples and the expectation $\mathbb{E}[\cdot]$ is over these samples. The same inequalities hold for $\mathcal{L} := \lambda^{-1} \mathcal{L}_{BA}$, with the identification $\epsilon = \lambda^{-1}$.*

### 4.4 Estimation of rate and distortion

Here, we describe our estimator for an upper bound $(\mathcal{D}, \mathcal{R})$ of $R(D)$ after solving the unconstrained problem (3). We provide more details in Sec. 8 of the Supplementary Material.

For any given pair of $\nu$ and $K$, we always have that $\mathcal{D} := \int \rho d(\mu \otimes K)$ and $\mathcal{R} := H(\mu \otimes K | \mu \otimes \nu)$ lie on an upper bound of $R(D)$ [Berger, 1971]. The two quantities can be estimated by standard Monte Carlo provided we can sample from $\mu \otimes K$ and evaluate the density $\frac{d\mu \otimes K}{d\mu \otimes \nu}(x, y) = \frac{dK(x, \cdot)}{d\nu}(y)$.

When only $\nu$ is given, e.g., obtained from optimizing (6) with WGD or NERD, we estimate an R-D upper bound as follows. As in the BA algorithm, we construct a kernel $K_\nu$ similarly to (11), i.e., $\frac{dK_\nu(x, \cdot)}{d\nu}(y) = \frac{e^{-\lambda \rho(x, y)}}{\int e^{-\lambda \rho(x, \tilde{y})} \nu(d\tilde{y})}$; then we estimate $(\mathcal{D}, \mathcal{R})$ using the pair $(\nu, K_\nu)$ as described earlier.

As NERD uses a continuous $\nu$, we follow [Lei et al., 2023a] and approximate it with its $n$-sample empirical measure to estimate $(\mathcal{D}, \mathcal{R})$. A limitation of NERD, BA, and our method is that they tend to converge to a rate estimate of at most $\log(n)$, where $n$ is the support size of $\nu$. This is because as the algorithms approach an $n$-point minimizer $\nu_n^*$ of the R-D problem, the rate estimate $\mathcal{R}$ approaches the mutual information of $\mu \otimes K_{\nu_n^*}$, which is upper-bounded by $\log(n)$ [Eckstein and Nutz, 2022]. In practice, this means if a target point of $R(D)$ has rate $r$, then we need $n \geq e^r$ to estimate it accurately.

### 4.5 Computational considerations

Common to all the aforementioned methods is the evaluation of a pairwise distortion matrix between $m$ points in $\mathcal{X}$ and $n$ points in $\mathcal{Y}$, which usually has a cost of $\mathcal{O}(mnd)$ for a $d$-dimensional source. While RD-VAE uses $n = 1$ (in the reparameterization trick), the other methods (BA, WGD, NERD) typically use a much larger $n$ and thus has the distortion computation as their main computation bottleneck. Compared to BA and WGD, the neural methods (RD-VAE, NERD) incur additional computation from neural network operations, which can be significant for large networks.

For NERD and WGD (and BA), the rate estimate upper bound of $\log(n)$ nats/sample (see Sec. 4.4) can present computational challenges. To target a high-rate setting, a large number of $\nu$ particles (high $n$) is required, and care needs to be taken to avoid running out of memory during the distortion matrix computation (one possibility is to use a small batch size $m$ with stochastic optimization).

## 5 Experiments

We compare the empirical performance of our proposed method (WGD) and its hybrid variant with Blahut–Arimoto (BA) [Blahut, 1972, Arimoto, 1972], RD-VAE [Yang and Mandt, 2022], and NERD [Lei et al., 2022] on the tasks of maximum-likelihood deconvolution and estimation of R-D upper bounds. While we experimented with WGD for both $\mathcal{L}_{BA}$ and $\mathcal{L}_{EOT}$, we found the former to be 10 to 100 times faster computationally while giving similar or better results; we therefore focus on WGD for $\mathcal{L}_{BA}$ in discussions below. For the neural-network baselines, we use the same (or as similar as possible) network architectures as in the original work [Yang and Mandt, 2022, Lei et al., 2023a]. We use the Adam optimizer for all gradient-based methods, except we use simple gradient descent

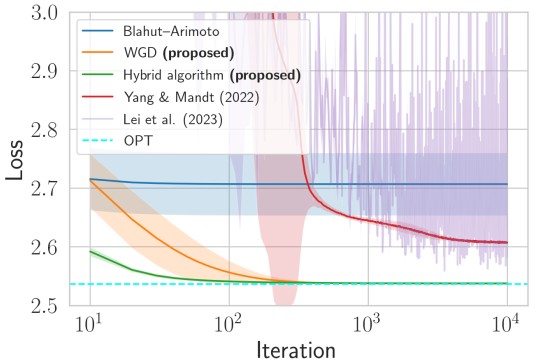

Figure 2: Losses over iterations. Shading corresponds to one standard deviation over random initializations.

Figure 3: Visualizing $\mu$ samples (top left), as well as the $\nu$ returned by various algorithms compared to the ground truth $\nu^*$ (cyan).

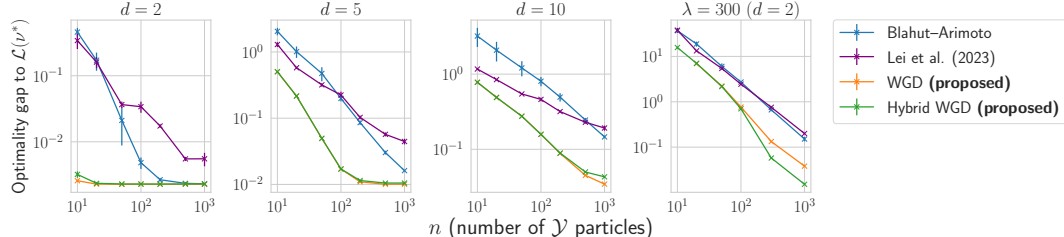

Figure 4: Optimality gap v.s. the number of particles $n$ used, on deconvolution problems with different dimension $d$ and distortion multiplier $\lambda$. The first three panels fix $\lambda = \sigma^{-2}$ and increase $d$, and the right-most panel corresponds to the 2-D problem with a higher $\lambda$ (denoising with a narrower Gaussian kernel). Overall the WGD methods attain higher accuracy for a given budget of $n$.

with a decaying step size in Sec. 5.1 to better compare the convergence speed of WGD and its hybrid variant. Further experiment details and results are given in the Supplementary Material Sec. 11.

## 5.1 Deconvolution

To better understand the behavior of the various algorithms, we apply them to a deconvolution problem with known ground truth (see Sec. 2.3). We adopt the Gaussian noise as before, letting $\alpha$ be the uniform measure on the unit circle in $\mathbb{R}^2$ and the source $\mu = \alpha * \mathcal{N}(0, \sigma^2)$ with $\sigma^2 = 0.1$.

We use $n = 20$ particles for BA, NERD, WGD and its hybrid variant. We use a two-layer network for NERD and RD-VAE with some hand-tuning (we replace the softplus activation in the original RD-VAE network by ReLU as it led to difficulty in optimization). Fig. 2 plots the resulting loss curves and shows that the proposed algorithms converge the fastest to the ground truth value $OPT := \mathcal{L}(\alpha)$. In Fig. 3, we visualize the final $\nu^{(t)}$ at the end of training, compared to the ground truth $\nu^* = \alpha$ supported on the circle (colored in cyan). Note that we initialize $\nu^{(0)}$ for BA, WGD, and its hybrid variant to the same $n$ random data samples. While BA is stuck with the randomly initialized particles and assigns large weights to those closer to the circle, WGD learns to move the particles to uniformly cover the circle. The hybrid algorithm, being able to reweight particles to reduce their transportation cost, learns a different solution where a cluster of particles covers the top-left portion of the circle with small weights while the remaining particles evenly covers the rest. Unlike our particle-based methods, the neural methods generally struggle to place the support of their $\nu$ exactly on the circle.

We additionally compare how the performance of BA, NERD, and the proposed algorithms scale to higher dimensions and a higher $\lambda$ (corresponding to lower entropic regularization in $\mathcal{L}_{EOT}$). Fig. 4 plots the gap between the converged and the optimal losses for the algorithms, and demonstrates the proposed algorithms to be more particle-efficient and scale more favorably than the alternatives which also use $n$ particles in the reproduction space. We additionally visualize how the converged particles for our methods vary across the R-D trade-off in Fig. 6 of the Supplementary Material.

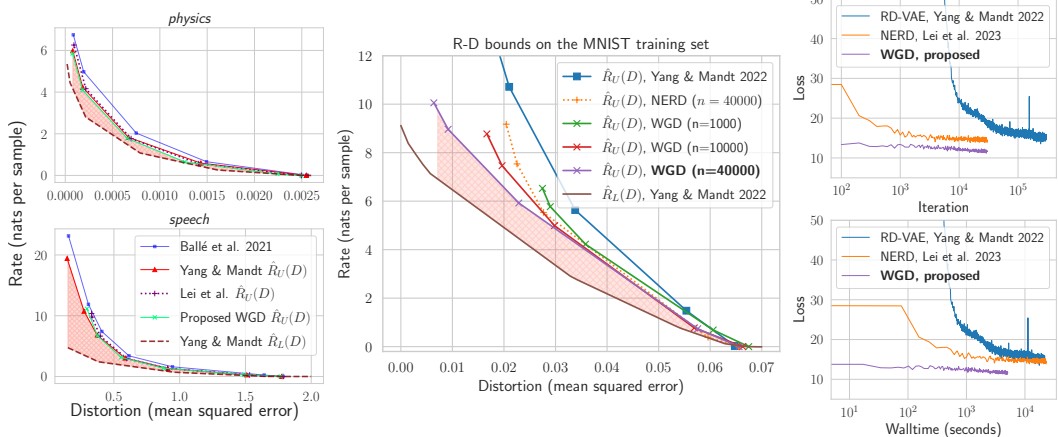

Figure 5: **Left, Middle:** R-D bound estimates on the physics, speech datasets [Yang and Mandt, 2022] and MNIST training set. **Right:** Example speed comparisons of WGD and neural upper bound methods on MNIST, with WGD converging at least an order of magnitude faster. On each of the dataset we also include an R-D *lower* bound estimated using the method of [Yang and Mandt, 2022].

## 5.2 Higher-dimensional data

We perform $R(D)$ estimation on higher-dimensional data, including the *physics* and *speech* datasets from [Yang and Mandt, 2022] and MNIST [LeCun et al., 1998]. As the memory cost to operating on the full datasets becomes prohibitive, we focus on NERD, RD-VAE, and WGD using mini-batch stochastic gradient descent. BA and hybrid WGD do not directly apply in the stochastic setting, as BA updates on random mini-batches can lead to divergence (as discussed in Sec. 4.2).

Fig. 5 plots the estimated R-D bounds on the datasets, and compares the convergence speed of WGD and neural methods in both iteration count and compute time. Overall, we find WGD to require minimal tuning and obtains the tightest R-D upper bounds within the $\log(n)$ rate limit (see Sec. 4.4), and consistently obtains tighter bounds than NERD given the same computation budget.

## 6 Discussions

In this work, we leverage tools from optimal transport to develop a new approach for estimating the rate-distortion function in the continuous setting. Compared to state-of-the-art neural approaches [Yang and Mandt, 2022, Lei et al., 2022], our Wasserstein gradient descent algorithm offers complementary strengths: 1) It requires a single main hyperparameter $n$ (the number of particles) and no network architecture tuning; and 2) empirically we found it to converge significantly faster and rarely end up in bad local optima; increasing $n$ almost always yielded an improvement (unless the bound is already close to being tight). From a modeling perspective, a particle representation may be inherently more efficient when the optimal reproduction distribution is singular or has many disconnected modes (shown, e.g., in Figs. 1 and 3). However, like NERD [Lei et al., 2022], our method has a fundamental limitation – it requires an $n$ that is exponential in the rate of the targeted $R(D)$ point to estimate it accurately (see Sec. 4.4). Thus, a neural method like RD-VAE [Yang and Mandt, 2022] may still be preferable on high-rate sources, while our particle-based method stands as a state-of-the-art solution on lower-rate sources with substantially less tuning or computation requirements.

Besides R-D estimation, our algorithm also applies to the mathematically equivalent problems of maximum likelihood deconvolution and projection under an entropic optimal transport (EOT) cost, and may find other connections and applications. Indeed, the EOT projection view of our algorithm is further related to optimization-based approaches to sampling [Wibisono, 2018], variational inference [Liu and Wang, 2016], and distribution compression [Shetty et al., 2021]. Our particle-based algorithm also generalizes optimal quantization (corresponding to $\epsilon = 0$ in $\mathcal{L}_{EOT}$ and projection of the source under the Wasserstein distance [Graf and Luschgy, 2007, Gray, 2013]) to incorporate a rate constraint ($\epsilon > 0$), and it would be interesting to explore the use of the resulting rate-distortion optimal quantizer for practical data compression and communication.

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

## Broader Impacts

Improved estimates on the fundamental cost of data compression aids the development and analysis of compression algorithms. This helps researchers and engineers make better decisions about where to allocate their resources to improve certain compression algorithms, and can translate to economic gains for the broader society. However, like most machine learning algorithms trained on data, the output of our estimator is only accurate insofar as the training data is representative of the population distribution of interest, and practitioners need to ensure this in the data collection process.

## Acknowledgements

We thank anonymous reviewers for feedback on the manuscript. Yibo Yang acknowledges support from the Hasso Plattner Foundation. Marcel Nutz acknowledges support from NSF Grants DMS-1812661, DMS-2106056. Stephan Mandt acknowledges support from the National Science Foundation (NSF) under the NSF CAREER Award 2047418; NSF Grants 2003237 and 2007719, the Department of Energy, Office of Science under grant DE-SC0022331, the IARPA WRIVA program, as well as gifts from Intel, Disney, and Qualcomm.

# Supplementary Material for
# Estimating the Rate-Distortion Function by Wasserstein Gradient Descent

We review probability theory background and explain our notation from the main text in Section 7, give the formulas we used for numerically estimating an $R(D)$ upper bound in Section 8, provide additional discussions and proofs regarding Wasserstein gradient descent in Section 9, elaborate on the connections between the R-D estimation problem and variational inference/learning in Section 10, provide additional experimental results and details in Section 11, and list an example implementation of WGD in Section 12. Our code and can be found at `https://github.com/yiboyang/wgd`.

## 7 Notions from probability theory

In this section we collect notions of probability theory used in the main text. See, e.g., [Çinlar, 2011] or [Folland, 1999] for more background.

**Marginal and conditional distributions.** The source and reproduction spaces $\mathcal{X}, \mathcal{Y}$ are equipped with sigma-algebras $\mathcal{A}_{\mathcal{X}}$ and $\mathcal{A}_{\mathcal{Y}}$, respectively. Let $\mathcal{X} \times \mathcal{Y}$ denote the product space equipped with the product sigma algebra $\mathcal{A}_{\mathcal{X}} \otimes \mathcal{A}_{\mathcal{Y}}$. For any probability measure $\pi$ on $\mathcal{X} \times \mathcal{Y}$, its first **marginal** is

$$\pi_1(A) := \pi(A \times \mathcal{Y}), \quad A \in \mathcal{A}_{\mathcal{X}},$$

which is a probability measure on $\mathcal{X}$. When $\pi$ is the distribution of a random vector $(X, Y)$, then $\pi_1$ is the distribution of $X$. The second marginal of $\pi$ is defined analogously as

$$\pi_2(B) := \pi(\mathcal{X} \times B), \quad B \in \mathcal{A}_{\mathcal{Y}}.$$

A Markov **kernel** or **conditional distribution** $K(x, dy)$ is a map $\mathcal{X} \times \mathcal{A}_{\mathcal{Y}} \to [0, 1]$ such that

1. $K(x, \cdot)$ is a probability measure on $\mathcal{Y}$ for each $x \in \mathcal{X}$;
2. the function $x \mapsto K(x, B)$ is measurable for each set $B \in \mathcal{A}_{\mathcal{Y}}$.

When speaking of the conditional distribution of a random variable $Y$ given another random variable $X$, we occasionally also use the notation $Q_{Y|X}$ from information theory [Polyanskiy and Wu, 2014]. Then, $Q_{Y|X=x}(B) = K(x, B)$ is the conditional probability of the event $\{Y \in B\}$ given $X = x$.

Suppose that a probability measure $\mu$ on $\mathcal{X}$ is given, in addition to a kernel $K(x, dy)$. Together they define a unique measure $\mu \otimes K$ on the product space $\mathcal{X} \times \mathcal{Y}$. For a rectangle set $A \times B \in \mathcal{A}_{\mathcal{X}} \otimes \mathcal{A}_{\mathcal{Y}}$,

$$\mu \otimes K(A \times B) = \int_A \mu(dx) K(x, B), \quad A \in \mathcal{A}_{\mathcal{X}}, B \in \mathcal{A}_{\mathcal{Y}}.$$

The measure $\pi := \mu \otimes K$ has first marginal $\pi_1 = \mu$.

The classic product measure is a special case of this construction. Namely, when a measure $\nu$ on $\mathcal{Y}$ is given, using the constant kernel $K(x, dy) := \nu(dy)$ (which does not depend on $x$) gives rise to the product measure $\mu \otimes \nu$,

$$\mu \otimes \nu(A \times B) = \mu(A)\nu(B), \quad A \in \mathcal{A}_{\mathcal{X}}, B \in \mathcal{A}_{\mathcal{Y}}.$$

Under mild conditions (for instance when $\mathcal{X}, \mathcal{Y}$ are Polish spaces equipped with their Borel sigma algebras, as in the main text), any probability measure $\pi$ on $\mathcal{X} \times \mathcal{Y}$ is of the above form. Namely, the **disintegration** theorem asserts that $\pi$ can be written as $\pi = \pi_1 \otimes K$ for some kernel $K$. When $\pi$ is the joint distribution of a random vector $(X, Y)$, this says that there is a measurable version of the conditional distribution $Q_{Y|X}$.

**Optimal transport.** Given a measure $\mu$ on $\mathcal{X}$ and a measurable function $T : \mathcal{X} \to \mathcal{Y}$, the **pushforward** (or image measure) of $\mu$ under $T$ is a measure on $\mathcal{Y}$, given by

$$T_{\#}\mu(B) = \mu(T^{-1}(B)), \quad B \in \mathcal{A}_{\mathcal{Y}}.$$

If $T$ is seen as a random variable and $\mu$ as the baseline probability measure, then $T_{\#}\mu$ is simply the distribution of $T$.

Suppose that $\mu$ and $\nu$ are probability measures on $\mathcal{X} = \mathcal{Y} = \mathbb{R}^d$ with finite second moment. As introduced in the main text, $\Pi(\mu, \nu)$ denotes the set of couplings, i.e., measures $\pi$ on $\mathcal{X} \times \mathcal{Y}$ with $\pi_1 = \mu$ and $\pi_2 = \nu$. The 2-Wasserstein distance $W_2(\mu, \nu)$ between $\mu$ and $\nu$ is defined as

$$W_2(\mu, \nu) = \left( \inf_{\pi \in \Pi(\mu,\nu)} \int \|y - x\|^2 \pi(dx, dy) \right)^{1/2}.$$

This indeed defines a metric on the space of probability measures with finite second moment.

We finish this section by giving a proof of the basic equivalence between optimization of $\mathcal{L}_{EOT}$ and $\mathcal{L}_{BA}$, which goes back to [Csiszár, 1974a, Lemma 1.3 and subsequent discussion]. Recall that $\Pi(\mu, \cdot)$ denotes the set of measures on the product space $\mathcal{X} \times \mathcal{Y}$ with $\pi_1 = \mu$.

**Lemma 7.1.** *Set $\epsilon = \lambda^{-1}$. It holds that*

$$\inf_{\nu \in \mathcal{P}(\mathcal{Y})} \mathcal{L}_{EOT}(\nu) = \inf_{\nu \in \mathcal{P}(\mathcal{Y})} \lambda^{-1} \mathcal{L}_{BA}(\nu) \quad \text{and} \quad \operatorname*{arg\,min}_{\nu \in \mathcal{P}(\mathcal{Y})} \mathcal{L}_{EOT}(\nu) = \operatorname*{arg\,min}_{\nu \in \mathcal{P}(\mathcal{Y})} \mathcal{L}_{BA}(\nu).$$

*Proof.* Both statements will follow from a simple property of relative entropy [Polyanskiy and Wu, 2022, Theorem 4.1, "golden formula"]: For $\pi \in \Pi(\mu, \cdot)$ with $H(\pi_2 \mid \nu) < \infty$, the properties of the logarithm reveal

$$H(\pi \mid \mu \otimes \nu) = H(\pi \mid \mu \otimes \pi_2) + H(\pi_2 \mid \nu), \tag{15}$$

and hence $H(\pi \mid \mu \otimes \nu) \geq H(\pi \mid \mu \otimes \pi_2)$. This implies

$$\inf_{\nu \in \mathcal{P}(\mathcal{Y})} \mathcal{L}_{EOT}(\nu) = \inf_{\pi \in \Pi(\mu,\cdot)} \int \rho \, d\pi + \epsilon H(\pi \mid \mu \otimes \pi_2)$$

$$= \inf_{\pi \in \Pi(\mu,\cdot)} \inf_{\nu \in \mathcal{P}(\mathcal{Y})} \int \rho \, d\pi + \epsilon H(\pi \mid \mu \otimes \nu)$$

$$= \frac{1}{\lambda} \inf_{\nu \in \mathcal{P}(\mathcal{Y})} \mathcal{L}_{BA}(\nu).$$

Further, any optimizer $\nu^*$ of $\mathcal{L}_{BA}$ with corresponding optimal "coupling" $\pi^* \in \Pi(\mu, \cdot)$ must satisfy $H(\pi_2^* \mid \nu^*) = 0$ (otherwise, taking $\nu^* = \pi_2^*$ has better objective) and thus $\pi_2^* = \nu^*$ and $\pi^* \in \Pi(\mu, \nu^*)$, therefore $\nu^*$ is also an optimizer of $\mathcal{L}_{EOT}$ by the above equality. Conversely, any optimizer $\nu^*$ of $\mathcal{L}_{EOT}$ with coupling $\pi^* \in \Pi(\mu, \nu^*) \subset \Pi(\mu, \cdot)$ clearly yields a feasible solution for $\mathcal{L}_{BA}$ as well, and hence is also an optimizer of $\mathcal{L}_{BA}$ by the above equality. $\qquad \square$

# 8  Numerical estimation of rate and distortion from a reproduction distribution

Given a reproduction distribution $\nu$ and a kernel $\mathcal{X} \times \mathcal{A}_{\mathcal{Y}} \to [0,1]$, the tuple $(\mathcal{D}, \mathcal{R}) \in \mathbb{R}^2$ defined by

$$\mathcal{D} := \int \rho \, d(\mu \otimes K)$$

and

$$\mathcal{R} := H(\mu \otimes K \mid \mu \otimes \nu)$$

lies above the $R(D)$ curve. This again follows from the variational inequality (15) $\mathcal{R} \geq H(\pi \mid \mu \times (\mu \times K)_2) = I(\mu \times K)$, so that $\mathcal{R} \geq R(\mathcal{D}) =: \inf_{\pi \in \Pi(\pi,\cdot):\pi(\rho) \leq \mathcal{D}} I(\pi)$.

Given only a reproduction distribution $\nu$, we will construct a kernel $K$ from $\nu$ and use $(\mu, K)$ to compute $(\mathcal{D}, \mathcal{R})$, letting

$$\frac{dK(x, \cdot)}{d\nu}(y) = \frac{e^{-\lambda \rho(x,y)}}{\int e^{-\lambda \rho(x,\tilde{y})} \nu(d\tilde{y})}$$

as in the first step of the BA algorithm. This choice of $K$ is in fact optimal as it achieves the minimum in the definition of $\mathcal{L}_{BA}$ (5) for the given $\nu$. Plugging $K$ into the formulas for $\mathcal{D}$ and $\mathcal{R}$ gives

$$\mathcal{D} = \int \rho(x,y)e^{\varphi(x)-\lambda\rho(x,y)}\mu\otimes\nu(dx,dy) \tag{16}$$

and

$$\mathcal{R} = \int\int \log\left(\frac{K(x,dy)}{\nu(dy)}\right)K(x,dy)\mu(dx) \tag{17}$$
$$= \int [\varphi(x)-\lambda\rho(x,y)]e^{\varphi(x)-\lambda\rho(x,y)}\mu\otimes\nu(dx,dy),$$

where we introduced the shorthand

$$\varphi(x) := -\log\int_{\mathcal{Y}} e^{-\lambda\rho(x,y)}\nu(dy). \tag{18}$$

Note that we have the following relation (which explains (6))

$$\mathcal{L}_{BA}(\nu) = \mathcal{R} + \lambda\mathcal{D} = \int \varphi(x)\mu(dx).$$

Let $\nu$ be an $n$-point measure, $\nu = \sum_{i=1}^{n} w_i\delta_{y_i}$, e.g., the output of WGD or in the inner step of NERD. Then $\varphi(x)$ can be evaluated exactly as a finite sum over the $\nu$ particles, and the expressions above for $\mathcal{D}$ and $\mathcal{R}$ (which are integrals w.r.t. the product distribution $\mu\otimes\nu$) can be estimated as sample averages. That is, given $m$ independent samples $\{x_i\}_{i=1}^{m}$ from $\mu$, we compute unbiased estimates

$$\hat{\mathcal{D}} = \sum_{i=1}^{m}\sum_{j=1}^{n}\frac{1}{m}w_j\rho(x_i,y_j)e^{\varphi(x_i)-\lambda\rho(x_i,y_j)},$$

$$\hat{\mathcal{R}} = \sum_{i=1}^{m}\sum_{j=1}^{n}\frac{1}{m}w_j[\varphi(x_i)-\lambda\rho(x_i,y_j)]e^{\varphi(x_i)-\lambda\rho(x_i,y_j)},$$

where $\varphi(x) = -\log\sum_{j=1}^{n} e^{\varphi(x_i)-\lambda\rho(x_i,y_j)}$. Similarly, a sample mean estimate for $\mathcal{L}_{BA}$ is given by

$$\hat{\mathcal{L}}_{BA}(\nu) = \frac{1}{m}\sum_{i=1}^{m}\varphi(x_i) = -\frac{1}{m}\sum_{i=1}^{m}\log\left(\sum_{j=1}^{n} e^{\varphi(x_i)-\lambda\rho(x_i,y_j)}\right). \tag{19}$$

In practice, we found it simpler and numerically more stable to instead compute $\hat{\mathcal{R}}$ as

$$\hat{\mathcal{R}} = \hat{\mathcal{L}}_{BA}(\nu) - \lambda\hat{\mathcal{D}}.$$

Whenever possible, we avoid exponentiation and instead use `logsumexp` to prevent numerical issues.

## 9 Wasserstein gradient descent

### 9.1 On Wasserstein gradients of the EOT and rate functionals

First, we elaborate on the Wasserstein gradient of the EOT functional $\mathcal{L}_{EOT}(\nu)$. That the dual potential from Sinkhorn's algorithm is differentiable follows from the fact that optimal dual potentials satisfy the Schrödinger equations (cf. [Nutz, 2021, Corollary 2.5]). Differentiability was shown in [Genevay et al., 2019, Theorem 2] in the compact setting, and in [Mena and Niles-Weed, 2019, Proposition 1] in unbounded settings. While Mena and Niles-Weed [2019] only states the result for quadratic cost, the approach of Proposition 1 therein applies more generally.

Below, we compute the Wasserstein gradient of the rate functional $\mathcal{L}_{BA}(\nu)$. Recall from (6),

$$\mathcal{L}_{BA}(\nu) = \int -\log\int \exp(-\lambda\rho(x,y))\nu(dy)\mu(dx).$$

Under sufficient integrability on $\mu$ and $\nu$ to exchange the order of limit and integral, we can calculate the first variation as

$$
\begin{aligned}
\lim_{\varepsilon \to 0} \frac{\mathcal{L}((1-\varepsilon)\nu + \varepsilon\tilde{\nu}) - \mathcal{L}(\nu)}{\varepsilon} &= -\int \lim_{\varepsilon \to 0} \frac{1}{\varepsilon} \log \left[ \frac{\int \exp(-\lambda\rho(x,y))(\nu + \varepsilon(\tilde{\nu} - \nu))(dy)}{\int \exp(-\lambda\rho(x,y))\nu(dy)} \right] \mu(dx) \\
&= -\int \lim_{\varepsilon \to 0} \frac{1}{\varepsilon} \log \left[ 1 + \frac{\int \exp(-\lambda\rho(x,y))\varepsilon(\tilde{\nu} - \nu)(dy)}{\int \exp(-\lambda\rho(x,y))\nu(dy)} \right] \mu(dx) \\
&= \iint -\frac{\exp(-\lambda\rho(x,y))}{\int \exp(-\lambda\rho(x,\tilde{y}))\nu(d\tilde{y})} \mu(dx) \, (\tilde{\nu} - \nu)(dy),
\end{aligned}
$$

where the last equality uses $\lim_{\varepsilon \to 0} \frac{1}{\varepsilon} \log(1 + \varepsilon x) = x$ and Fubini's theorem. Thus the first variation $\psi^\nu$ of $\mathcal{L}_{BA}$ at $\nu$ is

$$
\psi^\nu(y) = \int -\frac{\exp(-\lambda\rho(x,y))}{\int \exp(-\lambda\rho(x,\tilde{y}))\nu(d\tilde{y})} \mu(dx). \tag{20}
$$

To find the desired Wasserstein gradient of $\mathcal{L}_{BA}$, it remains to take the Euclidean gradient of $\psi^\nu$, i.e., $\nabla \mathcal{L}_{BA}(\nu) = \nabla \psi^\nu$. Doing so gives us the desired Wasserstein gradient:

$$
\begin{aligned}
\nabla V_{\mathcal{L}_{BA}}(\nu)[y] = \nabla_y \psi^\nu(y) &= \frac{\partial}{\partial y} \int -\frac{\exp(-\lambda\rho(x,y))}{\int \exp(-\lambda\rho(x,\tilde{y}))\nu(d\tilde{y})} \mu(dx) \\
&= \int \frac{\exp(-\lambda\rho(x,y))\lambda\frac{\partial}{\partial y}\rho(x,y)}{\int \exp(-\lambda\rho(x,\tilde{y}))\nu(d\tilde{y})} \mu(dx), \tag{21}
\end{aligned}
$$

again assuming suitable regularity conditions on $\rho$ and $\mu$ to exchange the order of integral and differentiation.

## 9.2 Proof of Proposition 4.2 (convergence of Wasserstein gradient descent)

We first provide an auxiliary result.

**Lemma 9.1.** *Let $\gamma_1 \geq \gamma_2 \geq \cdots \geq 0$ and $a_t \geq 0$, $t \in \mathbb{N}$, $C > 0$ satisfy $\sum_{t=1}^\infty \gamma_t = \infty$, $\sum_{t=1}^\infty \gamma_t^2 < \infty$, $\sum_{t=1}^\infty a_t \gamma_t < \infty$ and $|a_t - a_{t+1}| \leq C\gamma_t$ for all $t \in \mathbb{N}$. Then $\lim_{t\to\infty} a_t = 0$.*

*Proof.* The conclusion remains unchanged when rescaling $a_t$ by the constant $C$, and thus without loss of generality $C = 1$.

Clearly $\gamma_t \to 0$ as $\sum_{t=1}^\infty \gamma_t^2 < \infty$. Moreover, there exists a subsequence of $(a_t)_{t\in\mathbb{N}}$ which converges to zero (otherwise there exists $\delta > 0$ such that $a_t \geq \delta > 0$ for all but finitely many $t$, contradicting $\sum_{t=1}^\infty \gamma_t a_t < \infty$).

Arguing by contradiction, suppose that the conclusion fails, i.e., that there exists a subsequence of $(a_t)_{t\in\mathbb{N}}$ which is uniformly bounded away from zero, say $a_t \geq \delta > 0$ along that subsequence. Using this subsequence and the convergent subsequence mentioned above, we can construct a subsequence $a_{i_1}, a_{i_2}, a_{i_3}, \ldots$ where $a_{i_n} \approx 0$ for $n$ odd and $a_{i_n} \geq \delta$ for $n$ even. We will show that

$$
\sum_{t=i_{2n-1}}^{i_{2n}} a_t \gamma_t \gtrsim \delta^2/2 \qquad \text{for all } n \in \mathbb{N},
$$

contradicting the finiteness of $\sum_t \gamma_t a_t$. (The notation $\approx$ ($\gtrsim$) indicates (in)equality up to additive terms converging to zero for $n \to \infty$.)

To ease notation, fix $n$ and set $m = i_{2n-1}$ and $M = i_{2n}$. We show that $\sum_{t=m}^M a_t \gamma_t \gtrsim \delta^2/2$. To this end, using $|a_t - a_{t+1}| \leq \gamma_t$ we find

$$
a_t \geq a_M - \sum_{j=k}^{M-1} \gamma_j \geq \delta - \sum_{j=k}^{M-1} \gamma_j.
$$

Since $a_m \approx 0$, there exists a largest $n_0 \in \mathbb{N}$, $n_0 \geq m$, such that $\sum_{j=n_0}^{M-1} \gamma_j \gtrsim \delta$ (and thus $\sum_{j=n_0}^{M-1} \gamma_j \lesssim \delta - \gamma_{n_0} \approx \delta$ as well). We conclude

$$\sum_{t=m}^{M} \gamma_t a_t \geq \sum_{t=n_0}^{M} \gamma_t a_t \geq \sum_{t=n_0}^{M} \gamma_t \left( \delta - \sum_{j=k}^{M-1} \gamma_j \right) \gtrsim \delta^2 - \sum_{t=n_0}^{M} \sum_{j=n_0}^{M} \gamma_t \gamma_j \mathbf{1}_{\{j \geq k\}}$$

$$= \delta^2 - \frac{1}{2} \left( \sum_{t=n_0}^{M} \gamma_t \right)^2 - \frac{1}{2} \sum_{t=n_0}^{M} \gamma_t^2 \approx \delta^2/2,$$

where we used that $\sum_{t=n_0}^{M} \gamma_t^2 \approx 0$. This completes the proof. $\qquad\square$

*Proof of Proposition 4.2.* Using the linear approximation property in (14), we calculate

$$\mathcal{L}(\nu^{(n)}) - \mathcal{L}(\nu^{(0)}) = \sum_{t=0}^{n-1} \mathcal{L}(\nu^{(t+1)}) - \mathcal{L}(\nu^{(t)})$$

$$= \sum_{t=0}^{n-1} -\gamma_t \int \|\nabla V_{\mathcal{L}}(\nu^{(t)})\|^2 \, d\nu^{(t)} + \gamma_t^2 \, o\left( \int \|\nabla V_{\mathcal{L}}(\nu^{(t)})\|^2 \, d\nu^{(t)} \right).$$

As $\mathcal{L}(\nu^{(0)})$ is finite and $\mathcal{L}(\nu^{(n)})$ is bounded from below, it follows that

$$\sum_{t=0}^{\infty} \gamma_t \int \|\nabla V_{\mathcal{L}}(\nu^{(t)})\|^2 \, d\nu^{(t)} < \infty.$$

The claim now follow by applying Lemma 9.1 with $a_t = \int \|\nabla \psi^{\nu^{(t)}}\|^2 \, d\nu^{(t)}$; note that the assumption in the lemma, $|a_t - a_{t+1}| \leq C\gamma_t$, is satisfied due to the second inequality in (14) and a short calculation below

$$\left| \int \|\nabla \psi^{\nu^{(t)}}\|^2 - \int \|\nabla \psi^{\nu^{(t+1)}}\|^2 \right| \leq C W_2(\nu^{(t)}, \nu^{(t+1)})$$

$$\leq C \left( \int \int \|x - y\|^2 \delta_{x - \gamma_t \nabla V_{\mathcal{L}}(\nu^{(t)})[x]}(dy) \nu^{(t)}(dx) \right)^{\frac{1}{2}}$$

$$\leq C\gamma_t \left( \int \|\nabla V_{\mathcal{L}}(\nu^{(t)})\|^2 \nu^{(t)}(dx) \right)^{\frac{1}{2}}$$

$$\leq C\gamma_t \sup_{t'} \left( \int \|\nabla V_{\mathcal{L}}(\nu^{(t')})\|^2 \nu^{(t')}(dx) \right)^{\frac{1}{2}}$$

$$\leq C'\gamma_t,$$

where we use $\sup_t \int \|\nabla V_{\mathcal{L}}(\nu^{(t)})\|^2 \nu^{(t)}(dx) < \infty$ in the last step. $\qquad\square$

## 9.3 Proof of Proposition 4.3 (sample complexity)

Recall that $\mathcal{X} = \mathcal{Y} = \mathbb{R}^d$ and $\rho(x, y) = \|x - y\|^2$ in this proposition. For the proof, we will need the following lemma which is of independent interest. We write $\nu \leq_c \mu$ if $\nu$ is dominated by $\mu$ in convex order, i.e., $\int f \, d\nu \leq \int f \, d\mu$ for all convex functions $f : \mathbb{R}^d \to \mathbb{R}$.

**Lemma 9.2.** *Let $\mu$ have finite second moment. Given $\nu \in \mathcal{P}(\mathbb{R}^d)$, there exists $\tilde{\nu} \in \mathcal{P}(\mathbb{R}^d)$ with $\tilde{\nu} \leq_c \mu$ and*

$$\mathcal{L}_{EOT}(\mu, \tilde{\nu}) \leq \mathcal{L}_{EOT}(\mu, \nu).$$

*This inequality is strict if $\nu \not\leq_c \mu$. In particular, any optimizer $\nu^*$ of (8) satisfies $\nu^* \leq_c \mu$.*

*Proof.* Because this proof uses disintegration over $\mathcal{Y}$, it is convenient to reverse the order of the spaces in the notation and write a generic point as $(x, y) \in \mathcal{Y} \times \mathcal{X}$. Consider $\pi \in \Pi(\nu, \mu)$ and its disintegration $\pi = \nu(dx) \otimes K(x, dy)$ over $x \in \mathcal{Y}$. Define $T : \mathbb{R}^d \to \mathbb{R}^d$ by

$$T(x) := \int y \, K(x, dy).$$

Define also $\tilde{\pi} := (T, \mathrm{id})_{\#}\pi$ and $\tilde{\nu} := \tilde{\pi}_1$. From the definition of $T$, we see that $\tilde{\pi}$ is a martingale, thus $\tilde{\nu} \leq_c \mu$. Moreover, $\tilde{\nu} \otimes \mu = (T, \mathrm{id})_{\#}\nu \otimes \mu$. The data-processing inequality now shows that

$$H(\tilde{\pi}|\tilde{\nu} \otimes \mu) \leq H(\pi|\nu \otimes \mu).$$

On the other hand, $\int \|\int \tilde{y} K(x, d\tilde{y}) - y\|^2 K(x, dy) \leq \int \|x - y\|^2 K(x, dy)$ since the barycenter minimizes the squared distance, and this inequality is strict whenever $x \neq \int \tilde{y} K(x, d\tilde{y})$. Thus

$$\int \|x - y\|^2 \, \tilde{\pi}(dx, dy) \leq \int \|x - y\|^2 \, \pi(dx, dy),$$

and the inequality is strict unless $T(x) = x$ for $\nu$-a.e. $x$, which in turn is equivalent to $\pi$ being a martingale. The claims follow. $\qquad\square$

*Proof of Proposition 4.3.* Subgaussianity of the optimizer follows directly from Lemma 9.2.

Recalling that $\inf_\nu \mathcal{L}_{EOT}(\nu)$ and $\inf_\nu \lambda^{-1}\mathcal{L}_{BA}(\nu)$ have the same values and minimizers (given by (9) in Sec. 2.2), it suffices to show the claim for $\mathcal{L} = \mathcal{L}_{EOT}$. Let $\nu^*$ be an optimizer of (8) (i.e., an optimal reproduction distribution) and $\nu^n$ its empirical measure from $n$ samples, then clearly

$$\left| \min_{\nu_n \in \mathcal{P}_n(\mathbb{R}^d)} \mathcal{L}_{EOT}(\mu, \nu_n) - \min_{\nu \in \mathcal{P}(\mathbb{R}^d)} \mathcal{L}_{EOT}(\mu, \nu) \right| = \min_{\nu_n \in \mathcal{P}_n(\mathbb{R}^d)} \mathcal{L}_{EOT}(\mu, \nu_n) - \min_{\nu \in \mathcal{P}(\mathbb{R}^d)} \mathcal{L}_{EOT}(\mu, \nu)$$
$$\leq \mathbb{E}\left[ |\mathcal{L}_{EOT}(\mu, \nu^n) - \mathcal{L}_{EOT}(\mu, \nu^*)| \right]$$

where the expectation is taken over samples for $\nu^n$. The first inequality of Proposition 4.3 now follows from the sample complexity result for entropic optimal transport in [Mena and Niles-Weed, 2019, Theorem 2].

Denote by $\nu_m^*$ the optimizer for the problem (8) with $\mu$ replaced by $\mu^m$. Similarly to the above, we obtain

$$\mathbb{E}\left[ \left| \min_{\nu \in \mathcal{P}(\mathbb{R}^d)} \mathcal{L}_{EOT}(\mu, \nu) - \min_{\nu \in \mathcal{P}(\mathbb{R}^d)} \mathcal{L}_{EOT}(\mu^m, \nu) \right| \right]$$
$$\leq \mathbb{E}\left[ \max_{\nu \in \{\nu^*, \nu_m^*\}} |\mathcal{L}_{EOT}(\mu, \nu) - \mathcal{L}_{EOT}(\mu^m, \nu)| \right],$$

where the expectation is taken over samples from $\mu^m$. In this situation, we cannot directly apply [Mena and Niles-Weed, 2019, Theorem 2]. However, the bound given by [Mena and Niles-Weed, 2019, Proposition 2] still applies, and the only dependence on $\nu \in \{\nu^*, \nu_m^*\}$ is through their subgaussianity constants. By Lemma 9.2, these constants are bounded by the corresponding constants of $\mu$ and $\mu^m$. Thus, the arguments in the proof of [Mena and Niles-Weed, 2019, Theorem 2] can be applied, yielding the second inequality of Proposition 4.3.

The final inequality of Proposition 4.3 follows from the first two inequalities (the first one being applied with $\mu^m$) and the triangle inequality, where we again use the arguments in the proof of [Mena and Niles-Weed, 2019, Theorem 2] to bound the expectation over the subgaussianity constants of $\mu^m$. $\qquad\square$

## 9.4 Convergence of the proposed hybrid algorithm

In our present work, our hybrid algorithm targets the non-stochastic setting and is motivated by the fact that the BA update is in some sense orthogonal to the Wasserstein gradient update, and can only monotonically improve the objective. While empirically we observe the additional BA steps to not hurt – but rather accelerate – the convergence of WGD (see 5.1), additional effort is required to theoretically guarantee convergence of the hybrid algorithm.

There are two key properties we use for the convergence of the base WGD algorithm: 1) a certain monotonicity of the update steps (up to higher order terms, gradient descent improves the objective) and 2) stability of gradients across iterations. If we include the BA step, we find that 1) still holds, but 2) may a-priori be lost. Indeed, 1) holds since BA updates monotonically improve the objective. Using just 1), we can still obtain a Pareto convergence of the gradients for the hybrid algorithm, $\sum_{t=0}^{\infty} \gamma_t \int \|\nabla V_{\mathcal{L}}(\nu^{(t)})\|^2 \, d\nu^{(t)} < \infty$ (here $\nu^{(t)}$ are the outputs from the respective BA

Table 1: Guide to notation and their interpretations in various problem domains. "LVM" stands for latent variable modeling, "NPMLE" stands for non-parametric maximum-likelihood estimation. The R-D problem (3) is equivalent to a projection problem under an entropic optimal transport cost (discussed in Sec. 2.2) and statistical problems involving maximum-likelihood estimation (see discussion in Sec. 2.3 and below).

| Context | $\mu = P_X$ | $\rho(x, y)$ | $K = Q_{Y\|X}$ | $\nu = Q_Y$ |
|---|---|---|---|---|
| OT | source distribution | transport cost | "transport plan" | target distribution |
| R-D | data distribution | distortion criterion | compression algorithm | codebook distribution |
| LVM/NPMLE | data distribution | "$-\log p(x\|y)$" | variational posterior | prior distribution |
| deconvolution | noisy measurements | "noise kernel" | — | noiseless model |

steps and $\gamma_t$ is the step size of the gradient steps). Without property 2), we cannot conclude $\int \|\nabla V_{\mathcal{L}}(\nu^{(t)})\|^2 \, d\nu^{(t)} \to 0$ for $t \to \infty$. We emphasize that in practice, it still appears that 2) holds even after including the BA step. Motivated by this analysis, an adjusted hybrid algorithm where, e.g., the BA update is rejected if it causes a drastic change in the Wasserstein gradient, could guarantee that 2) holds with little practical changes. From a different perspective, we also believe the hybrid algorithm may be tractable to study in relation to gradient flows in the Wasserstein-Fisher-Rao geometry (cf. [Yan et al., 2023]), in which the BA step corresponds to a gradient update in the Fisher-Rao geometry with a unit step size.

In the stochastic setting, the BA (and therefore our hybrid) algorithm does not directly apply, as performing BA updates on mini-batches no longer guarantees monotonic improvement of the overall objective. Extending the BA and hybrid algorithm to this setting would be interesting future work.

## 10 R-D estimation and variational inference/learning

In this section, we elaborate on the connection between the R-D problem (3) and variational inference and learning in latent variable models, of which maximum likelihood deconvolution (discussed in Sec. 2.3) can be seen as a special case. Also see Section 3 of [Yang et al., 2023] for a related discussion in the context of lossy data compression.

To make the connections clearer to a general machine learning audience, we adopt notation more common in statistics and information theory. Table 1 summarizes the notation and the correspondence to the measure-theoretic notation used in the main text.

In statistics, a common goal is to model an (unknown) data generating distribution $P_X$ by some density model $\hat{p}(x)$. In particular, here we will choose $\hat{p}(x)$ to be a latent variable model, where $\mathcal{Y}$ takes on the role of a latent space, and $Q_Y = \nu$ is the distribution of a latent variable $Y$ (which may encapsulate the model parameters). As we shall see, the optimization objective defining the rate functional (5) corresponds to an aggregate Evidence LOwer Bound (ELBO) [Blei et al., 2017]. Thus, computing the rate functional corresponds to variational inference [Blei et al., 2017] in a given model (see Sec. 10.2), and the parametric R-D estimation problem, i.e.,

$$\inf_{\nu \in \mathcal{H}} \mathcal{L}_{BA}(\nu),$$

is equivalent to estimating a latent variable model using the variational EM algorithm [Beal and Ghahramani, 2003] (see Sec. 10.3). The variational EM algorithm can be seen as a restricted version of the BA algorithm (see Sec. 10.3), whereas the EM algorithm [Dempster et al., 1977] shares its E-step with the BA algorithm but can differ in its M-step (see Sec. 10.4).

### 10.1 Setup

For concreteness, fix a reference measure $\zeta$ on $\mathcal{Y}$, and suppose $Q_Y$ has density $q(y)$ w.r.t. $\zeta$. Often the latent space $\mathcal{Y}$ is a Euclidean space, and $q(y)$ is the usual probability density function w.r.t. the Lebesgue measure $\zeta$; or when the latent space is discrete/countable, $\zeta$ is the counting measure and $q(y)$ is a probability mass function. We will consider the typical parametric estimation problem and choose a particular parametric form for $Q_Y$ indexed by a parameter vector $\theta$. This defines a parametric family $\mathcal{H} = \{Q_Y^\theta : \theta \in \Theta\}$ for some parameter space $\Theta$. Finally, suppose the distortion

function $\rho$ induces a conditional likelihood density via $p(x|y) \propto e^{-\lambda \rho(x,y)}$, with a normalization constant that is constant in $y$.

These choices then result in a latent variable model specified by the joint density $q(y)p(x|y)$, and we model the data distribution with the marginal density: [2]

$$\hat{p}(x) = \int_{\mathcal{Y}} p(x|y)dQ_Y(y) = \int_{\mathcal{Y}} p(x|y)q(y)\zeta(dy). \tag{22}$$

As an example, a Gaussian mixture model with isotropic component variances can be specified as follows. Let $Q_Y$ be a mixing distribution on $\mathcal{X} = \mathcal{Y} = \mathbb{R}^d$ parameterized by component weights $w_{1,\dots,k}$ and locations $\mu_{1,\dots,k}$, such that $Q_Y = \sum_{k=1}^K w_k \delta_{\mu_k}$. Let $p(x|y) = \mathcal{N}(y, \sigma^2)$ be a conditional Gaussian density with mean $y$ and variance $\sigma^2$. Now formula (22) gives the usual Gaussian mixture density on $\mathbb{R}^d$.

Maximum-likelihood estimation then ideally maximizes the population log (marginal) likelihood,

$$\mathbb{E}_{x \sim P_X}[\log \hat{p}(x)] = \int \log \hat{p}(x) P_X(dx) = \int \log \left( \int_{\mathcal{Y}} p(x|y)dQ_Y(y) \right) P_X(dx). \tag{23}$$

The maximum-likelihood deconvolution setup can be seen as a special case where the form of the marginal density (22) derives from knowledge of the true data generation process, with $P_X = \alpha * \mathcal{N}(0, \frac{1}{\lambda})$ for some unknown $\alpha$ and known noise $\mathcal{N}(0, \frac{1}{\lambda})$ (i.e., the model is well-specified). We note in passing that the idea of estimating an unknown data distribution by adding artificial noise to it also underlies work on spread divergence [Zhang et al., 2020] and denoising score matching [Vincent, 2011].

To deal with the often intractable marginal likelihood in the inner integral, we turn to variational inference and learning [Jordan et al., 1999, Wainwright et al., 2008].

## 10.2 Connection to variational inference

Given a latent variable model and any data observation $x$, a central task in Bayesian statistics is to infer the Bayesian posterior [Jordan, 1999], which we formally view as a conditional distribution $Q^*_{Y|X=x}$. It is given by

$$\frac{dQ^*_{Y|X=x}(y)}{dQ_Y(y)} = \frac{p(x|y)}{\hat{p}(x)},$$

or, in terms of the density $q(y)$ of $Q_Y$, given by the following conditional density via the familiar Bayes' rule,

$$q^*(y|x) = \frac{p(x|y)q(y)}{\hat{p}(x)} = \frac{p(x|y)q(y)}{\int_{\mathcal{Y}} p(x|y)q(y)\zeta(dy)}.$$

Unfortunately, the true Bayesian posterior is typically intractable, as the (marginal) data likelihood in the denominator involves an often high-dimensional integral. Variational inference [Jordan et al., 1999, Wainwright et al., 2008] therefore aims to approximate the true posterior by a variational distribution $Q_{Y|X=x} \in \mathcal{P}(\mathcal{Y})$ by minimizing their relative divergence $H(Q_{Y|X=x}|Q^*_{Y|X=x})$. The problem is equivalent to maximizing the following lower bound on the marginal log-likelihood, known as the Evidence Lower BOund (ELBO) [Blei et al., 2017]:

$$\underset{Q_{Y|X=x}}{\arg\min} H(Q_{Y|X=x}|Q^*_{Y|X=x}) = \underset{Q_{Y|X=x}}{\arg\max} \text{ELBO}(Q_Y, x, Q_{Y|X=x}),$$

$$\text{ELBO}(Q_Y, x, Q_{Y|X=x}) = \mathbb{E}_{y \sim Q_{Y|X=x}}[\log p(x|y)] - H(Q_{Y|X=x}|Q_Y)$$

$$= \log \hat{p}(x) - H(Q_{Y|X=x}|Q^*_{Y|X=x}). \tag{24}$$

Recall the rate functional (5) arises through an optimization problem over a transition kernel $K$,

$$\mathcal{L}_{BA}(\nu) = \inf_K \lambda \int \rho d(\mu \otimes K) + H(\mu \otimes K | \mu \otimes \nu).$$

---

[2]To be more precise, we always fix a reference measure $\eta$ on $\mathcal{X}$, so that densities such as $\hat{p}(x)$ and $p(x|y)$ are with respect to $\eta$. In the applications we considered in this work, $\eta$ is the Lebesgue measure on $\mathcal{X} = \mathbb{R}^d$.

Translating the above into the present notation ($\mu \to P_X, K \to Q_{Y|X}, \nu \to Q_Y$; see Table 1), we obtain

$$\mathcal{L}_{BA}(Q_Y) = \inf_{Q_{Y|X}} \mathbb{E}_{x \sim P_X, y \sim Q_{Y|X=x}}[-\log p(x|y)] + \mathbb{E}_{x \sim P_X}[H(Q_{Y|X=x}|Q_Y)] + \text{const}$$
$$= \inf_{Q_{Y|X}} \mathbb{E}_{x \sim P_X}[-\text{ELBO}(Q_Y, x, Q_{Y|X=x})] + \text{const}. \tag{25}$$

which allows us to interpret the rate functional as the result of performing variational inference through ELBO optimization. At optimality, $Q_{Y|X} = Q^*_{Y|X}$, the ELBO (24) is tight and recovers $\log \hat{p}(x)$, and the rate functional takes on the form of a (negated) population marginal log likelihood (23), as given earlier by (6) in Sec. 2.1.

## 10.3 Connection to variational EM

The discussion so far concerns *probabilistic inference*, where a latent variable model $(Q_Y, p(x|y))$ has been given and we saw that computing the rate functional amounts to variational inference. Suppose now we wish to *learn* a model from data. The R-D problem (4) then corresponds to model estimation using the variational EM algorithm [Beal and Ghahramani, 2003].

To estimate a latent variable model by (approximate) maximum-likelihood, the variational EM algorithm maximizes the population ELBO

$$\mathbb{E}_{x \sim P_X}[\text{ELBO}(Q_Y, x, Q_{Y|X=x})] = \mathbb{E}_{x \sim P_X, y \sim Q_{Y|X=x}}[\log p(x|y)] - \mathbb{E}_{x \sim P_X}[H(Q_{Y|X=x}|Q_Y)], \tag{26}$$

w.r.t. $Q_Y$ and $Q_{Y|X}$. This precisely corresponds to the R-D problem $\inf_{Q_Y \in \mathcal{H}} \mathcal{L}_{BA}(Q_Y)$, using the form of $\mathcal{L}_{BA}(Q_Y)$ from (25).

In popular implementations of variational EM such as the VAE [Kingma and Welling, 2013], $Q_Y$ and $Q_{Y|X}$ are restricted to parametric families. When they are allowed to range over all of $\mathcal{P}(\mathcal{Y})$ and all conditional distributions, variational EM then becomes equivalent to the BA algorithm.

## 10.4 The Blahut–Arimoto and (exact) EM algorithms

Here we focus on the distinction between the BA algorithm and the (exact) EM algorithm [Dempster et al., 1977], rather than the *variational EM* algorithm. Both BA and (exact) EM perform coordinate descent on the same objective function (namely the negative of the population ELBO from (26)), but they define the coordinates slightly differently — the BA algorithm uses $(Q_{Y|X}, Q_Y)$ with $Q_Y \in \mathcal{P}(\mathcal{Y})$, whereas the EM algorithm uses $(Q_{Y|X}, \theta)$ with $\theta$ indexing a parametric family $\mathcal{H} = \{Q_Y^\theta : \theta \in \Theta\}$. Thus the coordinate update w.r.t. $Q_{Y|X}$ (the "E-step") is the same in both algorithms, but the subseuqent "M-step" potentially differs depending on the role of $\theta$.

Given the optimization objective, which is simply the negative of (26),

$$\mathbb{E}_{x \sim P_X, y \sim Q_{Y|X=x}}[-\log p(x|y)] + H(P_X Q_{Y|X}|P_X \otimes Q_Y), \tag{27}$$

both BA and EM optimize the transition kernel $Q_{Y|X}$ the same way in the E-step, as

$$\frac{dQ^*_{Y|X=x}}{dQ_Y}(y) = \frac{p(x|y)}{\hat{p}(x)}. \tag{28}$$

For the M-step, the BA algorithm only minimizes the relative entropy term of the objective (27),

$$\min_{Q_Y \in \mathcal{P}(\mathcal{Y})} H(P_X Q^*_{Y|X}; P_X \otimes Q_Y),$$

(with the optimal $Q_Y$ given by the second marginal of $P_X Q^*_{Y|X}$) whereas the EM algorithm minimizes the full objective w.r.t. the parameters $\theta$ of $Q_Y$,

$$\min_{\theta \in \Theta} \mathbb{E}_{(x,y) \sim P_X Q^*_{Y|X}}[-\log p(x|y)] + H(P_X Q^*_{Y|X}; P_X \otimes Q_Y). \tag{29}$$

The difference comes from the fact that when we parameterize $Q_Y$ by $\theta$ in the parameter estimation problem, $Q^*_{Y|X}$ — and consequently both terms in the objective of (29) — will have functional dependence on $\theta$ through the E-step optimality condition (28).

In the Gaussian mixture example, $Q_Y = \sum_{k=1}^{K} w_k \delta_{\mu_k}$, and its parameters $\theta$ consist of the components weights $(w_1, ..., w_K) \in \Delta^{d-1}$ and location vectors $\{\mu_1, ..., \mu_K\} \subset \mathbb{R}^d$. The E-step computes $Q_{Y|X=x}^* = \sum_k w_k \frac{p(x|\mu_k)}{p(x)} \delta_{\mu_k}$. For the M-step, if we regard the locations as known so that $\theta = (w_1, ..., w_K)$ only consists of the weights, then the two algorithms perform the same update; however if $\theta$ also includes the locations, then the M-step of the EM algorithm will not only update the weights as in the BA algorithm, but also the locations, due to the distortion term $\mathbb{E}_{(x,y) \sim P_X Q_{Y|X}^*}[-\log p(x|y)] = -\int \sum_k w_k \frac{p(x|\mu_k)}{p(x)} \log p(x|\mu_k) P_X(dx)$.

# 11 Further experimental results

Our deconvolution experiments were run on Intel(R) Xeon(R) CPUs, and the rest of the experiments were run on Titan RTX GPUs.

In most experiments, we use the Adam [Kingma and Ba, 2015] optimizer for updating the $\nu$ particle locations in WGD and for updating the variational parameters in other methods. For our hybrid WGD algorithm, which adjusts the particle weights in addition to their locations, we found that applying momentum to the particle locations can in fact slow down convergence, and therefore use plain gradient descent with a decaying step size.

To trace an R-D upper bound for a given source, we solve the unconstrained R-D problem 3 for a heuristically-chosen grid of $\lambda$ values similarly to those in [Yang and Mandt, 2022], e.g., on a log-linear grid $\{1e4, 3e3, 1e3, 3e2, 1e2, ...\}$. Alternatively, a constrained optimization method like the Modified Differential Multiplier Method (MDMM) [Platt and Barr, 1987] can be adopted to directly target $R(D)$ for various values of $D$s. The latter approach will also help us identify when we run into the $\log(n)$ rate limit of particle-based methods (Sec. 4.4): suppose we perform constrained optimization with a distortion threshold of $D$; if the chosen $n$ is not large enough, i.e., $\log(n) < R(D)$, then no $\nu$ supported on (at most) $n$ points is feasible for the given constraint (otherwise we have a contradiction). When this happens, the Lagrange multiplier associated with the infeasiable constraint ($\lambda$ in our case) will be observed to increase indefinitely rather than stabalize to a finite value under a method like MDMM.

## 11.1 Deconvolution

**Architectures for the neural baselines** For the RD-VAE, we used a similar architecture as the one used on the banana-shaped source in [Yang and Mandt, 2022], consisting of two-layer MLPs for the encoder and decoder networks, and Masked Autoregressive Flow [Papamakarios et al., 2017] for the variational prior. For NERD, we follow similar architecture settings as [Lei et al., 2023a], using a two-layer MLP for the decoder network. Following Yang and Mandt [2022], we initially used softplus activation for the MLP, but found it made the optimization difficult; therefore we switched to ReLU activation which gave much better results. We conjecture that the softplus activation led to overly smooth mappings, which had difficulty matching the optimal $\nu^* = \alpha$ measure which is concentrated on the unit circle and does not have a Lebesgue density.

**Experiment setup** As discussed in Sec. 4.2, BA and our hybrid algorithms do not apply to the stochastic setting; to be able to include them in our comparison, the input to all the algorithms is an empirical measure $\mu^m$ (training distribution) with $m = 100000$ samples, given the same fixed seed. We found the number of training samples is sufficiently large that the losses do not differ significantly on the training distribution v.s. random/held-out samples.

Recall from Sec. 2.3, given data observations following $\mu = \alpha * \mathcal{N}(0, \sigma^2)$, if we perform deconvolution with an assumed noise variance $\frac{1}{\lambda}$ for some $\frac{1}{\lambda} \leq \sigma^2$, then the optimal solution to the problem is given by $\nu^* = \alpha_\lambda = \alpha * \mathcal{N}(0, \frac{1}{\lambda})$. We compute the optimal loss $OPT = \mathcal{L}_{BA}(\nu^*)$ by a Monte-Carlo estimate, using formula (19) with $m = 10^4$ samples from $\mu$ and $n = 10^6$ samples from $\nu^*$. The resulting $\hat{\mathcal{L}}_{BA}$ is positively biased (it overestimates the $OPT$ in expectation) due to the bias of the plug-in estimator for $\varphi(x)$ (18), so we use a large $n$ to reduce this bias. [3] Note the same issue occurs in the NERD algorithm (13).

---

[3]Another, more sophisticated solution would be annealed importance sampling [Grosse et al., 2015] or a related method developed for estimating marginal likelihoods and partition functions.

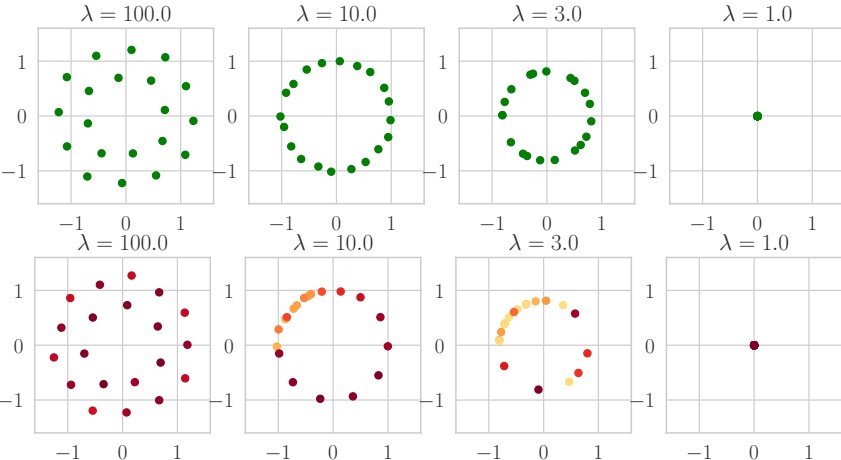

Figure 6: Visualizing the converged $n = 20$ particles of WGD (top row) and hybrid WGD (bottom row) estimated on $m = 10000$ source samples in the deconvolution problem (Sec. 5.1), for decreasing distortion penalty $\lambda$. The case $\lambda = 10.0 = \sigma^{-2}$ corresponds to "complete noise removal" of the source and recovers the ground truth $\alpha$ (unit circle).

**Optimized reproduction distributions** To visualize the (continuous) $\nu$ learned by RD-VAE and NERD, we fit a kernel density estimator on 10000 $\nu$ samples drawn from each, using `seaborn.kdeplot`.

In Figure 6, we provide additional visualization for the particles estimated from the training samples by WGD and its hybrid variant, for a decreasing distortion penalty $\lambda$ (equivalently, increasing entropic regularization strength $\epsilon$).

## 11.2   Higher-dimensional datasets

For NERD, we set $n$ to the default $40000$ in the code provided by [Lei et al., 2023a], on all three datasets.

For WGD, we used $n = 4000$ on *physics*, 200000 on *speech*, and 40000 on MNIST (see also smaller $n$ for comparison in Fig. 5).

On *speech*, both NERD and WGD encountered the issue of a $\log(n)$ upper bound on the rate estimate as described in Sec. 4.4. Therefore, we increased $n$ to 200000 for WGD and obtain a tighter R-D upper bound than NERD, particularly in the low-distortion regime. Such a large $n$ incurred out-of-memory error for NERD.

We borrow the R-D upper and lower bound results of [Yang and Mandt, 2022] from their paper on *physics* and *speech*. We obtain sandwich bounds on MNIST using a similar neural network architecture and other hyperparameter choices as in the GAN-generated image experiments of [Yang and Mandt, 2022] (using a ResNet-VAE for the upper bound and a CNN for the lower bound), except we set a larger $k = 10000$ in the lower bound experiment; we suspect the resulting lower bound is still far from being tight.

## 12   Example implementation of WGD

We provide a self-contained minimal implementation of Wasserstein gradient descent on $\mathcal{L}_{BA}$, using the Jax library [Bradbury et al., 2018]. To compute the Wasserstein gradient, we evaluate the first variation of the rate functional in `compute_psi_sum` according to (20), yielding $\sum_{i=1}^{n} \psi^{\nu}(x_i)$, then simply take its gradient w.r.t. the particle locations $x_{1,\dots n}$ using Jax's autodiff tool on line 51.

The implementation of WGD on $\mathcal{L}_{EOT}$ is similar, except the first variation is computed using Sinkhorn's algorithm. Both versions can be easily just-in-time compiled and enjoy GPU acceleration.

```
1   # Wasserstein GD on the rate functional L_{BA}.
2   import jax.numpy as jnp
3   import jax
4   from jax.scipy.special import logsumexp
5
6   # Define the distortion function \rho.
7   squared_diff = lambda x, y: jnp.sum((x - y) ** 2)
8   pairwise_distortion_fn = jax.vmap(jax.vmap(squared_diff, (None, 0)), (0, None))
9
10
11  def wgrad(mu_x, mu_w, nu_x, nu_w, rd_lambda):
12      """
13      Compute the Wasserstein gradient of the rate functional, which we will use
14      to move the \nu particles.
15      :param mu_x: locations of \mu atoms.
16      :param mu_w: weights  of \mu atoms.
17      :param nu_x: locations of \nu atoms.
18      :param nu_w: weights  of \nu atoms.
19      :param rd_lambda: R-D tradeoff hyperparameter.
20      :return:
21      """
22
23      def compute_psi_sum(nu_x):
24          """
25          Here we compute a surrogate loss based on the first variation \psi, which
26          allows jax autodiff to compute the desired Wasserstein gradient.
27          :param nu_x:
28          :return: psi_sum = \sum_i \psi(nu_x[i])
29          """
30          C = pairwise_distortion_fn(mu_x, nu_x)
31          scaled_C = rd_lambda * C   # [m, n]
32          log_nu_w = jnp.log(nu_w)   # [1, n]
33
34          # Solve BA inner problem with a fixed nu.
35          phi = - logsumexp(-scaled_C + log_nu_w, axis=1, keepdims=True)   # [m, 1]
36          loss = jnp.sum(mu_w * phi)   # Evaluate the rate functional. Eq (6) in paper.
37
38          # Let's also report rate and distortion estimates (discussed in Sec. 4.4 of the paper).
39          # Find \pi^* via \phi
40          pi = jnp.exp(phi - scaled_C) * jnp.outer(mu_w, nu_w)   # [m, n]
41          distortion = jnp.sum(pi * C)
42          rate = loss - rd_lambda * distortion
43
44          # Now evaluate \psi on the atoms of \nu.
45          phi = jax.lax.stop_gradient(phi)
46          psi = - jnp.sum(jnp.exp(jax.lax.stop_gradient(phi) - scaled_C) * mu_w, axis=0)
47          psi_sum = jnp.sum(psi)   # For computing gradient w.r.t. each nu_x atom.
48          return psi_sum, (loss, rate, distortion)
49
50      # Evaluate the Wasserstein gradient, i.e., \nabla \psi, on nu_x.
51      psi_prime, loss = jax.grad(compute_psi_sum, has_aux=True)(nu_x)
52      return psi_prime, loss
53
54
55  def wgd(X, n, rd_lambda, num_steps, lr, rng):
56      """
57      A basic demo of Wasserstein gradient descent on a discrete distribution.
58      :param X: a 2D array [N, d] of data points defining the source \mu.
59      :param n: the number of particles to use for \nu.
```

```python
60      :param rd_lambda: R-D tradeoff hyperparameter.
61      :param num_steps: total number of gradient updates.
62      :param lr:  step size.
63      :param rng: jax random key.
64      :return: (nu_x, nu_w), the locations and weights of the final \nu.
65      """
66      # Set up the source measure \mu.
67      m = jnp.size(X, 0)
68      mu_x = X
69      mu_w = 1 / m * jnp.ones((m, 1))
70      # Initialize \nu atoms using random training samples.
71      rand_idx = jax.random.permutation(rng, m)[:n]
72      nu_x = X[rand_idx]   # Locations of \nu atoms.
73      nu_w = 1 / n * jnp.ones((1, n))   # Uniform weights.
74      for step in range(num_steps):
75          psi_prime, (loss, rate, distortion) = wgrad(mu_x, mu_w, nu_x, nu_w, rd_lambda)
76          nu_x -= lr * psi_prime
77          print(f'step={step}, loss={loss:.4g}, rate={rate:.4g}, distortion={distortion:.4g}')
78
79      return nu_x, nu_w
80
81
82  if __name__ == '__main__':
83      # Run a toy example on 2D Gaussian samples.
84      rng = jax.random.PRNGKey(0)
85      X = jax.random.normal(rng, [10, 2])
86      nu_x, nu_w = wgd(X, n=4, rd_lambda=2., num_steps=100, lr=0.1, rng=rng)
```

