# OpenReview forum: "Estimating the Rate-Distortion Function by Wasserstein Gradient Descent"
_NeurIPS.cc/2023/Conference — NeurIPS 2023 poster_

### Official Review · Reviewer_RV2s · 2023-06-25

**Soundness:** 3 good
**Presentation:** 3 good
**Contribution:** 3 good
**Rating:** 7
**Confidence:** 5

**Summary:**

This paper describes the elegant connection between entropic OT and estimation of rate-distortion functions, as well as proposes a novel algorithm based on Wasserstein Gradient Descent (WGD). It outlines prior methods (e.g. Blahut-Arimoto, NERD) in the same (or similar) language, and performs WGD on the two objective functions of interest, detailing the gradient calculations, and provides numerical experiments comparing the methods to prior works in the literature. In these experiments, we see that WGD is able to provide good bounds on the rate distortion function


**Strengths:**

This paper wrote down a very elegant connection between rate distortions + entropic optimal transport -- as someone who studies the latter, I think this is great. I think it's surprising this connection has not been made explicit between the two communities. Their proposed method is also quite insightful. Numerical and statistical results are convincing as well.

**Weaknesses:**

There is minimal technical novelty in this paper, though I think that's completely fine given the newfound connections the authors make bridging two academic communities. I think the references are a bit thin, especially on the level of the Wasserstein gradient for $\mathcal{L}_{EOT}$. A recent paper, for example, is [Rigollet2022Sample], as well as some of the papers it cites, and the papers that cite it.

@article{rigollet2022sample,
  title={On the sample complexity of entropic optimal transport},
  author={Rigollet, Philippe and Stromme, Austin J},
  journal={arXiv preprint arXiv:2206.13472},
  year={2022}
}

It is also worth pointing out that this constant "$C$" in (15) of the text scales exponentially poorly with the regularization parameter $\epsilon$.

**Questions:**

Would it be interesting to analyze the "debiased" entropic OT dynamics for this problem as in e.g. [Pooladian2022Debiaser] and [Feydy2019Interpolating]?

@inproceedings{pooladian2022debiaser,
  title={Debiaser beware: Pitfalls of centering regularized transport maps},
  author={Pooladian, Aram-Alexandre and Cuturi, Marco and Niles-Weed, Jonathan},
  booktitle={International Conference on Machine Learning},
  pages={17830--17847},
  year={2022},
  organization={PMLR}
}

@inproceedings{feydy2019interpolating,
  title={Interpolating between optimal transport and mmd using sinkhorn divergences},
  author={Feydy, Jean and S{\'e}journ{\'e}, Thibault and Vialard, Fran{\c{c}}ois-Xavier and Amari, Shun-ichi and Trouv{\'e}, Alain and Peyr{\'e}, Gabriel},
  booktitle={The 22nd International Conference on Artificial Intelligence and Statistics},
  pages={2681--2690},
  year={2019},
  organization={PMLR}
}


Minor comments:

L33: “R-D” function instead of $R(D)$ (L86, L99, L101, L110, …) [just to be consistent]

L221: “optimize”

Missing \mathbb{E} in the statement of Proposition 4.3? You also could write this as one line, saving a lot of space (since the rates are basically the same for all the statements).


**Limitations:**

Yes

---

> ### Author Rebuttal · Authors · 2023-08-09
>
> Thank you for the detailed and insightful comments.  We have corrected all the typographical issues as you suggested, and address all your remaining concerns and questions below.
>
> > I think the references are a bit thin, especially on the level of the Wasserstein gradient for $\mathcal{L}\_{EOT}$.
>
> Thank you for pointing us to the more recent references, which we have included in our revision. Particularly, we added more recent papers like [Chizat 2022] and [Yan et al. 2023] in relation to gradient flows for $\mathcal{L}\_{EOT}$ and $\mathcal{L}\_{BA}$ . We further added references to [Genevay et al, 2019] and [Rigollet and Stromme, 2022] in relation to the sample complexity result. The result of [Rigollet and Stromme, 2022] is particularly interesting because it could be used to derive a version of our Proposition 4.3 with distortion functions other than the quadratic.
>
> > It is also worth pointing out that this constant "C" in (15) of the text scales exponentially poorly with the regularization parameter
>
> This is true. However, unlike the case where one uses EOT as an approximation of OT, in the compression problem we are not only interested in the small regularization regime. The setting of low rate and high distortion (corresponding to large entropic regularization) has received increasing interest from the generative modeling community [Mentzer et al., 2020; Yang et al., 2023], where techniques such as GANs and diffusion models allow compression algorithms to achieve realistic image reconstructions with extremely low bit-rates, at the cost of a large squared-error distortion. In that sense, the poor scaling of C is less problematic for R-D estimation than for other applications of EOT.
>
>
> > Questions:
> > Would it be interesting to analyze the "debiased" entropic OT?
>
> We focus on the “biased” entropic OT distance since it has the exact mathematical form of the lossy compression / R-D problem. The “debiased” entropic OT distances may also admit information-theoretic interpretations and would be interesting to investigate in future work.
>
>
> > Missing \mathbb{E} in the statement of Proposition 4.3?
>
> The first line of Proposition 4.3 does not have an expectation sign because there the LHS is already deterministic.  The following lines involve the $m$-sample empirical measure $\mu^m$, which is random and we take expectation over the samples.
>
>
> ----
> **References**
>
> Lénaïc Chizat. Mean-field langevin dynamics: Exponential convergence and annealing. arXiv:2202.01009, 2022
>
> Yuling Yan, Kaizheng Wang, and Philippe Rigollet. Learning gaussian mixtures using the wasserstein-fisher-rao gradient flow. arXiv:2301.01766, 2023
>
> Aude Genevay, Lénaic Chizat, Francis Bach, Marco Cuturi, and Gabriel Peyré. Sample complexity of Sinkhorn divergences. AISTATS, 2019
>
> Philippe Rigollet and Austin J Stromme. On the sample complexity of entropic optimal transport. arXiv:2206.13472, 2022
>
> Mentzer, F., Toderici, G. D., Tschannen, M., and Agustsson, E. High-fidelity generative image compression. Advances in Neural Information Processing Systems, 33:11913–11924, 2020.
>
> Yang, Y., Mandt, S., and Theis, L. An introduction to neural data compression. arXiv:2202.06533,2022

---

### Official Review · Reviewer_4bjw · 2023-06-26

**Soundness:** 3 good
**Presentation:** 3 good
**Contribution:** 3 good
**Rating:** 7
**Confidence:** 3

**Summary:**

The authors propose a novel method based on Wasserstein Gradient Descent for numerically computing the Rate-Distortion function. Judging from the experiments their method seems to achieve comparable performance to other state of the art methods while having lower computational cost and being conceptually simpler, which reduces the need for tuning (of neural network architectures).

**Strengths:**

- The method is novel and well motivated
- Good choice of numerical experiments

**Weaknesses:**

Lack of theoretical analysis/guarantees  (see also questions)

**Questions:**

Some questions on the theory side:
- In (15) in line 202, should there be an inequality between the first and second line?
- Should the integral in Lemma 4.2 not be w.r.t \nu^t and not \nu^k ?
- I am not quite sure I correctly understand the implications of Proposition 4.3.
How can anything be implied about WGD if we do not know that WGD converges to a global optimizer i.e. how does WGD relate to \min considered in the bound of Proposition 4.3 ?
To me right now it looks like a theoretical heuristic.

Small Typos:
- Line 172: baised should be biased
- Line 221: optimie should be optimize
- Line 302: Should "articles" be particles?
- Line 389: One of the authors is shown as Le?nai?c  (not sure if this might be caused on my end)

**Limitations:**

It would be interesting to have more theoretical guarantees for this method and/or more experiments on other realistic data.

---

> ### Author Rebuttal · Authors · 2023-08-09
>
> Thank you for the insightful review. We have incorporated all the typographical/notational suggestions as well as added a new experiment on MNIST (see PDF attached to the main rebuttal). Below we go over all the concerns and how we addressed them in our updated manuscript.
>
> > In (15) in line 202, should there be an inequality between the first and second line?
>
> There should have been a comma; fixed.
>
> > Should the integral in Lemma 4.2 not be w.r.t \nu^t and not \nu^k ?
>
> Yes; fixed.
>
> > I am not quite sure I correctly understand the implications of Proposition 4.3. How can anything be implied about WGD if we do not know that WGD converges to a global optimizer i.e. how does WGD relate to \min considered in the bound of Proposition 4.3 ? To me right now it looks like a theoretical heuristic.
>
> Proposition 4.3 is a statement about the approximation capabilities of the extremum estimator $\min\_{\nu \in \mathcal{P}\_n} \mathcal{L}(\nu)$ rather than the actual algorithm for computing it. We have now made this aspect clearer in the updated manuscript.
>
> In statistics, it is common to consider the theoretical merit (e.g., consistency, bias) of an estimator, such as the MLE, separately from the computational procedure for obtaining it; similarly, machine learning often studies the approximation ability of a hypothesis class (such as neural networks) separately from the training problem (e.g., the numerics of SGD). Our Proposition 4.3 is a statement of the former kind. It assures the consistency of the R-D estimator $\min_{\nu \in \mathcal{P}_n} \mathcal{L}(\nu)$ (defined by "the best R-D loss achievable over all reproduction measures supported on at most $n$ points") and further quantifies its rate of convergence in terms of: 1). the number of source samples ($m$), characterizing its statistical efficiency; and 2). the number of particles used ($n$), which implies a statement of universal approximation analogous to that of neural networks. While the *asymptotic* consistency of such an R-D estimator has been shown in information theory literature [Harrison and Kontoyiannis, 2008], we are not aware of any previous *finite-sample* bounds like ours.
>
> ----
> **References**
>
> Matthew T. Harrison and Ioannis Kontoyiannis. Estimation of the rate–distortion function. IEEE Transactions on Information Theory, 54(8):3757–3762, 2008.

---

> > ### Comment · Reviewer_4bjw · 2023-08-17
> >
> > I would like to thank the authors for their clear response to my questions which have addressed my concerns. In accordance I have slightly increased the evaluation.

---

### Official Review · Reviewer_GB3T · 2023-07-01

**Soundness:** 2 fair
**Presentation:** 1 poor
**Contribution:** 3 good
**Rating:** 5
**Confidence:** 3

**Summary:**

The paper proposes an algorithm for estimating the rate-distortion function for (possibly) continuous sources. This is done by presenting the R-D as an entropy-regularized optimal transport problem, and solving via Wasserstein GD. The source distribution is approximated by empirical distributions. This yields an upper bound on the true R,D. Some latest results from EOT are adapted in order to bound the theoretical deviation from the real R-D for sub-Gaussian sources. The convergence of the method is demonstrated through simulations.

**Strengths:**

The proposed method seems novel, and theoretical contribution about sample complexity (Prop. 4.3) seems important.

**Weaknesses:**

My concerns here are twofold:

1.	First, the proposed Algorithm outputs the marginal distribution of the reproduction. This allows to compute an upper bound for the R-D function (as described in Sec. 4.4), but only where the Blahut-Arimoto step (11) is tractable, which is a major drawback. It might also incur in an additional error (beyond the result of Prop. 4.3), which is not discussed.

2.	Second, and more crucial, Regarding the clarity of the text itself. Writing is *very* unclear and not well-organized. Just to name a few issues: The definition for the Wasserstein gradient is not clear, and it is unclear how to compute it. $\psi^t$ is used sometimes for a gradient, and sometimes for a potential (whose calculation is also not explained). Eq. (15) is missing some relation sign (probably $\leq$). In Lemma 4.2, $k$ is not defined (a typo, maybe?). Many cases of "it is well known" or "we know" without any reference. Many more typos. I'm afraid this does not meet the quality requirements for NeuRIPS.


**Questions:**

The paper contains a decent theoretical contribution, but the writing needs a total makeover. I would recommend an acceptance only if this issue could be fixed for the final version, which I doubt.

**Limitations:**

The Algorithm outputs only the marginal distribution of the reproduction and provides an upper bound for the R-D function only where the Blahut-Arimoto step is tractable. Neither this, nor other limitations, are discussed.

---

> ### Author Rebuttal · Authors · 2023-08-09
>
> Thank you for your helpful suggestions, which have significantly improved our manuscript. Below we go over your concerns and detail how we have addressed them in our updated manuscript.
>
> > First, the proposed Algorithm outputs the marginal distribution of the reproduction. This allows to compute an upper bound for the R-D function (as described in Sec. 4.4), but only where the Blahut-Arimoto step (11) is tractable, which is a major drawback. It might also incur in an additional error (beyond the result of Prop. 4.3), which is not discussed.
>
> Our original writing appears to have been misleading; we have rephrased it to emphasize that the upper bound computation in our method is in fact tractable, and there is no additional error incurred. While the BA-like step (11) involves an integral w.r.t. the $\nu$ distribution, in our algorithm $\nu$ is represented as $n$ particles, so the integral reduces to a finite sum, and this step is tractable and no more expensive than one update in the WGD algorithm.  Our original writing on line 242 (“provided the integral in the numerator can be computed exactly”) referred to the general setting which applied to both our method and NERD. Indeed, NERD uses a continuous $\nu$ and requires additional approximation for this computation (as explained on lines 244-248).
>
>
> > Second, and more crucial, Regarding the clarity of the text itself. Writing is very unclear and not well-organized...
>
> We sincerely apologize for the unclear writing, numerous typos and poor notation in our rushed write-up ahead of the submission deadline.  We have since made extensive revisions to improve the writing and correct all errors. Going over the list of issues (also see overall response #1, 2):
> > The definition for the Wasserstein gradient is not clear, and it is unclear how to compute it.
>
> We revised our introduction of the Wasserstein gradient, separating the formal definition (Definition 4.1; as the Euclidean gradient of the first variation) from its linearization property (eq 15). We explain that the former gives us the computational recipe, while the latter characterization justifies its role as a bona fide gradient (i.e., “taking a small enough step along the Wasserstein gradient indeed decreases the loss functional”). We add references to [Ambrosio et al. 2008, Definition 10.1.1], [Chizat 2022, Lemma A.2] and [Carlier et al. 2022, Proposition 4.2] to precisely relate the definition and the linearization property in our setting.
>
> > $\psi^t$ is used sometimes for a gradient, and sometimes for a potential (whose calculation is also not explained).
>
> This was an unfortunate notation clash; we have now changed the notation for the Wasserstein gradient to $\Psi$. We improved the discussion around its calculation based on Def 4.1, clarifying that it comes from a straightforward differentiation of the Sinkhorn potential (i.e. the output of Sinkhorn’s algorithm) in the case of $\mathcal{L}\_{EOT}$, and we added in Section 4.1 an explicit formula in the case of $\mathcal{L}\_{BA}$  (originally given by eq. 23 in the Supplementary Material).
>
> > Eq. (15) is missing some relation sign (probably).
>
> We now added the missing comma.
> > In Lemma 4.2, k is not defined (a typo, maybe?).
>
> Typo indeed, corrected as $t$.
>
> > Many cases of "it is well known" or "we know" without any reference.
>
> We have eliminated all instances of "it is well known" or "we know" and replaced them with detailed references.
>
> **References**
>
> Lénaïc Chizat. Mean-field langevin dynamics: Exponential convergence and annealing. arXiv:2202.01009, 2022
>
> Luigi Ambrosio, Nicola Gigli, and Giuseppe Savaré. Gradient flows in metric spaces and in the space of probability measures.
>
> Guillaume Carlier, Lénaïc Chizat, and Maxime Laborde. Lipschitz continuity of the Schrödinger map in entropic optimal transport. arXiv:2210.00225, 2022

---

> > ### Comment · Reviewer_GB3T · 2023-08-12
> >
> > I would like to thank the authors for their response and clarifications. I will recommend acceptance, but still insist that published version should be more readable.

---

> > > ### Author Response · Authors · 2023-08-16
> > > **Thank you for the positive re-evaluation; let us know how we address any remaining concerns**
> > >
> > > We would like to thank reviewer **GB3T** for their positive re-evaluation. We believe the reviewer was mainly concerned about
> > > 1. the tractability of our upper bound estimate;
> > > 2. unclear presentation, especially the technical discussion around the Wasserstein gradient.
> > >
> > > On #1, we hope our explanation about the upper bound estimate (see above and global rebuttal point 1) clarified the reviewer's concern about its tractability. Let us know otherwise and we are happy to discuss it further.
> > >
> > > One #2, we strongly agree with improving the readability of the writing prior to publication, and have therefore made extensive revisions according to reviewers' suggestions (listed above and outlined in the global response).
> > >
> > > Please let us know if you have additional suggestions or how we can address your remaining concerns.

---

### Official Review · Reviewer_SDqV · 2023-07-07

**Soundness:** 3 good
**Presentation:** 2 fair
**Contribution:** 3 good
**Rating:** 8
**Confidence:** 2

**Summary:**

This paper proposes estimating the rate distortion function using Wesserstein gradient descent. Different from the classical Blahut-Arimoto algorithm in which the support points are fixed, the proposed method is able to learn the support of the optimal distribution. The authors prove finite-sample complexity bounds, and also conduct experiments demonstrating the superiority of the proposed method.

**Strengths:**

1. By drawing interesting connections between the R-D problem and entropic optimal transport, this paper proposes a novel algorithm which is conceptually simple but effective.

2. The authors combine the advantage of Wasserstein gradient descent with BA and come up with a hybrid method that can update both the particle weights and the support of the distribution.

3. In the experiments section, the authors make a rigorous comparision of the proposed method with various existing methods, from the classic BA algorithm to modern neural network based approaches.

**Weaknesses:**

While this paper is technically solid, some parts of the paper is hard to read. For example $I(X;Y)$ in eq.(1) is not defined, and the boundeness assumption in Lemma 4.2 my need additional explanation.

**Questions:**

1. As the authors mention in Sec. 4.2, the hybrid algorithm alternates between WGD and BA. While this seems like a heuristics-based method, would this hurt the convergence properties of WGD?

2. Lemma 4.2 only implies that WGD would converge to stationary points. Is it possible that under suitable conditions on the objective function, it will converge to minimizers?

**Limitations:**

The authors have adequately addressed the limitations and potential negative societal impact.

---

> ### Author Rebuttal · Authors · 2023-08-09
>
> Thank you for your helpful feedback. We have incorporated your suggestions to improve the writing.  Below we will go over all points raised and detail how they have been addressed.
>
> > Weaknesses:
> > While this paper is technically solid, some parts of the paper is hard to read. For example
> I(X;Y) in eq.(1) is not defined, and the boundeness assumption in Lemma 4.2 my need additional explanation.
>
> $I(X;Y)$ stands for the mutual information of the joint distribution $P_X Q_{Y|X}$, which we now define after eq (1) in our updated manuscript.
> We have also clarified the boundedness assumption in Lemma 4.2, specifically the condition $\sup_t \int \| \nabla V_{\mathcal{L}}(\nu^{(t)})\|^2 \,d\nu^{(t)} \leq \infty$.  We replaced “assume that” with “suppose that”, to make it clear that this is the hypothesis of the lemma. This condition is satisfied when, e.g., the cost function $\rho$ has a bounded derivative on the relevant domain, which is the case in our examples.
>
> > Questions:
> > As the authors mention in Sec. 4.2, the hybrid algorithm alternates between WGD and BA. While this seems like a heuristics-based method, would this hurt the convergence properties of WGD?
>
> This is an interesting question, and one we can only partly answer.
> The hybrid algorithm is motivated by the fact that the BA update is in some sense orthogonal to the Wasserstein gradient update, and can only monotonically improve the objective.  While empirically we observe the BA steps to not hurt -- but rather accelerate -- the convergence of WGD (see Section 5.2), additional effort is required to theoretically guarantee convergence of the hybrid algorithm.
>
> There are two key properties we use for the convergence of WGD: 1) a certain monotonicity of the update steps (up to higher order terms, gradient descent improves the objective) and 2) stability of gradients across iterations. If we include the BA step, we find that 1) still holds, but 2) may a-priori be lost. Indeed, 1) holds since BA updates monotonically improve the objective. Using just 1), we can still obtain a Pareto convergence of the gradients for the hybrid algorithm, $\sum_{t=0}^\infty \gamma_t \int \|\nabla V_{\mathcal{L}}(\nu^{(t)})\|^2 \,d\nu^{(t)} < \infty$ (here $\nu^{(t)}$ are the outputs from the respective BA steps and $\gamma_t$ is the step size of the gradient steps). Without property 2), we cannot conclude $\int \|\nabla V_{\mathcal{L}}(\nu^{(t)})\|^2 \,d\nu^{(t)} \rightarrow 0$ for $t\rightarrow \infty$. We emphasize that in practice, it still appears that 2) holds even after including the BA step. Motivated by this analysis, an adjusted hybrid algorithm where, e.g., the BA update is rejected if it causes a drastic change in the Wasserstein gradient, could guarantee that 2) holds with little practical changes. From a different perspective, we also believe the hybrid algorithm may be tractable to study in relation to gradient flows in the Wasserstein-Fisher-Rao geometry (cf. Yan et al. 2023). We have added a discussion in Section 4.2, and leave further investigation to future work.
>
>
> > Lemma 4.2 only implies that WGD would converge to stationary points. Is it possible that under suitable conditions on the objective function, it will converge to minimizers?
>
> As our objective is non-convex over its domain (probability measures supported on at most $n$ points), we could only show convergence to stationary points. This is a very common limitation in the literature on gradient descent algorithms. It is in contrast to the discrete R-D problem on a fixed finite alphabet, which is convex, finite-dimensional, and solvable (in principle) by the BA algorithm.
> It is theoretically possible for our algorithm to get stuck at a non-minimizer, but we have yet to observe this on real-world problems.
> It is worth mentioning here that in the infinite particle limit, Yan et al. 2023 recently proved that Wasserstein-Fisher-Rao gradient flow for the BA functional can only converge to a global minimum (if it converges). This is important insofar as our hybrid algorithm could be interpreted as an implementation of gradient descent in this Wasserstein-Fisher-Rao geometry. We discuss this in the revised version at the end of Section 3.

---

> > ### Comment · Reviewer_SDqV · 2023-08-18
> > **Response**
> >
> > I would like to thank the authors for their detailed response. I will keep my rating and still recommend accept.

---

### Official Review · Reviewer_jdNJ · 2023-07-08

**Soundness:** 4 excellent
**Presentation:** 3 good
**Contribution:** 2 fair
**Rating:** 5
**Confidence:** 5

**Summary:**

This paper proposes an estimator using WGD and moving particles. This is different from prior methods that leverage neural networks to fit the unknown high-dimensional support of the optimal reconstruction distribution. The authors also note a connection with entropic OT and provide sample complexity bounds on estimating R(D). Numerical results show that it provides tighter estimates of R(D) compared to prior estimators on high-dimensional data.

**Strengths:**

- The method proposed is very principled, and well presented. The ideas are clear and the use of particles is natural for this problem. Showing how it can be implemented in practice was quite interesting.
- The finite sample complexity result is nice, as it is the first such result in this line of work. Additionally, it shows one interesting use of the connection between R(D) and entropic OT. It would also be good to cite other recent work that discuss this connection (https://arxiv.org/pdf/2212.10098.pdf, https://arxiv.org/pdf/2307.00246.pdf).
- The experimental results show good support of the method, with regards to the tightness of the bounds.

**Weaknesses:**

- While interesting, the work appears to be fairly incremental in terms of rate-distortion estimation. It provides slightly tighter bounds but fails to estimate at rates larger than log(n) which is a fundamental problem in rate-distortion (or mutual information) estimation (see https://arxiv.org/abs/1811.04251). Hence, it is difficult to say that this paper provides a wholly new solution to the rate-distortion estimation problem, as it seems to provide a third alternative to the same rate-distortion objective. In practice the prior work by Yang and Mandt is the only one so far applicable to most real-world data like images, where the rate per sample is far larger than log(n) for any feasible n.
- The computational (specifically, memory) cost of the particle method was not discussed, and this should be mentioned, especially if each particle consists of high-dimension vectors. The benefit of the neural methods is that one does not need to maintain many high dimensional vectors.

**Questions:**

- Do the neural methods (NERD and Yang and Mandt) provide tighter bounds as the network complexity or architecture increases? If complex enough, could they be comparable with the particle method as well?

**Limitations:**

Yes

---

> ### Author Rebuttal · Authors · 2023-08-09
>
> Thank you for your insightful feedback and bringing our attention to recent related work. We have followed your suggestion to cite (Wu et al. 2022, Lei et al. 2023), and added a new experiment on MNIST as well as more discussion around the computational cost of our method; also see points 2, 3, 5 of our main rebuttal.  Below we will go over all concerns raised and detail how they have been addressed.
>
> > While interesting, the work appears to be fairly incremental in terms of rate-distortion estimation. It provides slightly tighter bounds but fails to estimate at rates larger than log(n) which is a fundamental problem in rate-distortion (or mutual information) estimation…
>
> We agree that the log(n) issue is fundamental to all sample-based estimators of mutual information (and R-D), including ours and NERD, and neural methods are more suitable in certain regimes, e.g., for smooth, high-dimensional distributions. On the other hand, our results demonstrate that directly optimizing over particles can be more effective, for instance in settings where the optimal reproduction distribution is concentrated on low dimensional manifolds and/or have many widely-separated modes. This gives our method several distinct advantages over neural approaches:
> 1. **Ease of use for practitioners.** Our method completely removes the need to specify neural net architectures, and involves only choosing 1). a learning rate schedule (which we automate with adaptive methods like Adam) and 2). the number of particles $n$, which controls the solution quality. In our experience, the neural methods also tend to be harder to train: NERD requires experimenting with $n$ to ensure accurate gradient estimation (see the failure case in the left panel of Fig. 1 of the main text), and numerical issues from deep neural density models (deep normalizing flows) can also cause the training of RD-VAE to diverge.
>
> 2. **Improved R-D bounds with lower computation complexity.** Without much tuning, our method consistently produces R-D bounds comparable with or better than NERD, as well as RD-VAE up to the rate limit of log(n). At the same approximation quality, our method also tends to be more memory and computationally efficient in our experiments. We give more insight into this in our next response on computation efficiency.
>
> Finally, we want to highlight another aspect of our contribution to R-D theory, which is the introduction of a new, rich class of examples with known ground truth as a benchmark for algorithms. We discuss it more prominently in our updated manuscript, as explained in point # 5 of our main rebuttal.
>
> > The computational (specifically, memory) cost of the particle method was not discussed, and this should be mentioned, especially if each particle consists of high-dimension vectors. The benefit of the neural methods is that one does not need to maintain many high dimensional vectors.
>
> We now added a discussion in Section 4 about the computational complexity of WGD vs. the neural methods. Although WGD maintains potentially high-dimensional vectors, its per-iteration complexity (both in computation and memory) is largely comparable to that of NERD given the same $n$; this is because in both methods, the main bottleneck is in computing a pairwise distortion matrix whose cost scales as $O(m n d)$. Depending on the size of the generator network, NERD can be computationally more expensive than WGD, as it additionally requires backpropagation through the network.   RD-VAE also computes a distortion matrix but essentially with $n=1$; however its computation and memory requirements are dominated by that of training (typically large) neural density estimators.
> As different network architectures are used on different data sources (with varying levels of GPU support for operations such as convolution), it can be difficult to compare the computational complexity of WGD with the neural methods; nonetheless, we make a timing comparison in Fig 2 of the main rebuttal PDF, where the per-iteration time on MNIST is roughly 1:1.4:5 for WGD:RD-VAE:NERD.
> Finally, we remark that maintaining particles is inherently more efficient in the low distortion setting where the source consists of distinct atoms; in this regime, the neural methods also have to memorize the training data, but can only store it indirectly in the network parameters.
>
>
> > Do the neural methods (NERD and Yang and Mandt) provide tighter bounds as the network complexity or architecture increases? If complex enough, could they be comparable with the particle method as well?
>
> In theory, yes, all three methods (NERD, RD-VAE, and ours) converge to the true R-D in the ideal limit of infinite capacity / number of particles & samples. However, in practice it is a different question how efficiently each method estimates R-D given a computation budget. In our experience, increasing the network complexity for the neural methods can help produce tighter bounds, but this does not seem to happen consistently for various reasons.
>
> - On many experiments, the neural architectures we borrowed from the original papers appear to have been well optimized, and tweaking them yielded little or no gains (e.g., the DCGAN architecture for NERD on MNIST, or the hierarchical VAE for RD-VAE).
> - Increasing the network complexity may also make the training more difficult or even diverge, which we observed with RD-VAE.
> - In settings where the optimal reproduction distribution is concentrated on low dimensional manifolds and/or have many widely-separated modes, the neural approach is inherently less efficient because of the built-in bias of neural nets towards learning smooth mappings. The right-hand panel in Fig 1 shows one example where the optimal $\nu$ is concentrated on the unit circle and is recovered by WGD, while the neural methods produce $\nu$ smeared around the circle.
> By contrast, we observe consistently improving R-D estimates when increasing the number of particles in our approach.

---

> > ### Comment · Reviewer_jdNJ · 2023-08-21
> >
> > I thank the authors for their detailed response, and recommend acceptance for the paper.

---

### Author Rebuttal · Authors · 2023-08-09

We thank all reviewers for taking the time to review this manuscript and for providing insightful feedback, which has strengthened our submission.

We appreciate that the reviewers recognized our proposed method as “principled” (**jdNJ**), “novel” (**SDqV, GB3T, 4bjw, RV2s**), and “technically solid” (**jdNJ, SDqV, GB3T, RV2s**), and found our numerical results convincing (**jdNJ, SDqV, 4bjw, RV2s**). Moreover, our sample complexity result based on entropic optimal transport (Prop 4.3) “is the first such result in this line of work” (**jdNJ**) and “important” (**GB3T**).

The reviewers also pointed to some parts of our manuscript that were unclear, particularly the tractability of our upper bound estimate and the computational aspects of the Wasserstein gradient. We have taken the suggestions seriously and improved our manuscript accordingly. As we are not given the option to share our updated manuscript, we summarize the main improvements below:

1. [in response to **GB3T**] We have addressed the confusion around line 242 and made it clear that the upper bound from our method is in fact tractable. While the BA-like step (11) involves an integral w.r.t. the $\nu$ distribution, in our algorithm $\nu$ is represented as $n$ particles, so the integral reduces to a finite sum and this step is no more expensive than one update in the WGD algorithm.  Our original writing on line 242 referred to the general setting which applied to both our method and NERD. Indeed, NERD uses a continuous $\nu$ and requires additional approximation for this computation (as explained in lines 244-248).
2. [in response to **GB3T**] We have extensively revised the discussion around WGD to make it clearer and more accessible.
    - In our updated version, we now preface the definition of the Wasserstein gradient with the intuition of gradient flow in the space of probability measures.
    - We added a new explanation that the formal definition (as the Euclidean gradient of the first variation of the loss functional) is the computational basis of the algorithm, whereas the linearization property of the Wasserstein gradient in eq 15 justifies its role as a bona fide gradient and is what allows us to prove convergence of WGD to a stationary point (Lemma 4.2). We added detailed references to [Ambrosio et al. 2008, Definition 10.1.1], [Chizat 2022, Lemma A.2] and [Carlier et al. 2022, Proposition 4.2] to precisely relate the definition and the linearization property in our setting.
   - On the computation of the Wasserstein gradient (WG): for the EOT functional, we now state clearly that its WG is computed by taking the Euclidean gradient of the output of the Sinkhorn algorithm.  For the BA functional, the WG has an explicit formula (derived in Supplementary Material eq 23) which we now include in Section 4.1 of the main text.

3. [in response to **jdNJ, RV2s**] We expanded our references to include related work on the statistical complexity of EOT (Genevay et al. 2019, Riollet and Stromme 2022), non-parametric Gaussian mixture estimation (Yan et al., 2023), as well as recent papers in rate-distortion theory (Wu et al. 2022, Lei et al. 2023) which also note a connection between R-D and EOT. Wu et al. study the finite (and known) alphabet setting, similarly to the BA algorithm, and Lei et al. discuss the relation to optimal scalar quantization. By contrast, our work solves the computational problem of R-D estimation with Wasserstein gradient descent, with a focus on the continuous, high-dimensional, and stochastic optimization setting. Besides our methodological contribution, we also connect literature on related problems in statistics, information theory, and entropic optimal transport, and actually leverage the latter connection to characterize the sample complexity and approximation quality of our R-D estimator.

4. [in response to **4bjw**] To demonstrate the effectiveness and scalability of our method on more real-world data, we added a new experiment on the MNIST dataset. Again we obtain tighter R-D bounds with less computation than alternative methods. **Please see the attached PDF**.

5. [in response to **jdNJ**] One contribution to the R-D literature, which was not sufficiently emphasized in our first manuscript, is that we introduce a new, rich class of examples with known ground truth as a benchmark for algorithms. While the R-D literature usually uses Gaussian, Laplacian, or Bernoulli sources for that purpose, we leverage the connection with maximum likelihood deconvolution to find that a Gaussian convolution of *any* distribution can serve as a source with a known optimal reproduction distribution. While we already used this result to assess the optimality of various algorithms on the circular source in Section 5.2, we now discuss it more prominently in the updated manuscript.

----
**References**

Lénaïc Chizat. Mean-field langevin dynamics: Exponential convergence and annealing. arXiv:2202.01009, 2022

Luigi Ambrosio, Nicola Gigli, and Giuseppe Savaré. Gradient flows in metric spaces and in the space of probability measures.

Guillaume Carlier, Lénaïc Chizat, and Maxime Laborde. Lipschitz continuity of the Schrödinger map in entropic optimal transport. arXiv:2210.00225, 2022

Aude Genevay, Lénaic Chizat, Francis Bach, Marco Cuturi, and Gabriel Peyré. Sample complexity of Sinkhorn divergences. AISTATS, 2019

Philippe Rigollet and Austin J Stromme. On the sample complexity of entropic optimal transport. arXiv:2206.13472, 2022


Yuling Yan, Kaizheng Wang, and Philippe Rigollet. Learning gaussian mixtures using the wasserstein-fisher-rao gradient flow. arXiv:2301.01766, 2023

Shitong Wu, Wenhao Ye, Hao Wu, Huihui Wu, Wenyi Zhang, and Bo Bai. A communication optimal transport approach to the computation of rate distortion functions. arXiv:2212.10098, 2022

Eric Lei, Hamed Hassani, and Shirin Saeedi Bidokhti. On a relation between the rate-distortion function and optimal transport. arXiv:2307.00246, 2023

---

> ### Comment · Area_Chair_Pv2H · 2023-08-13
> **Follow-up questions**
>
> Dear Authors,
>
> Thank you very much for your response and clarifications!
> As AC, I would like to ask follow-up questions that allow me to better understand your contributions.
> The proposed algorithm (WGD) is similar to particle gradient descent introduced by [Nitanda, Atsushi, and Taiji Suzuki] which basically is gradient descent on the points in the support. In the specific regime of stepsizes $\gamma_k \to 0$, I think the two algorithms are equivalent. If this holds, then the convergence result in Lemma 4.2 can be recovered by invoking the convergence of gradient descent on smooth functions. Would you mind elaborating on this connection?
>
> Best regards,
> AC

---

> > ### Author Response · Authors · 2023-08-16
> > **Thank you for a thought-provoking question**
> >
> > Dear AC, thank you for your question. We are assuming that you are referring to ["Stochastic Gradient Descent for Infinite Ensembles (2017)"](https://arxiv.org/pdf/1712.05438.pdf) which we will abbreviate [NS].
> >
> > We agree there is an interesting but subtle connection, and there are a few aspects to this:
> >
> > 1. At a high level, the idea in [NS] is to consider a functional on the space of probability measures and evolve a cloud of particles towards a minimizer by following a kind of gradient with respect to a certain metric.  This idea is indeed the same as in our manuscript (and also many other papers).
> >
> > 2. Moreover, [NS] found that when using uniformly-weighted particles, the gradient descent algorithm in the space of probability measures (algorithm 1 of [NS]) becomes equivalent to Euclidean gradient descent of the loss function w.r.t. the particle locations (algorithm 3 of [NS]). We found this equivalence to also hold for our WGD algorithm *when optimizing the BA functional*, with the BA functional $\mathcal{L}\_{BA}$ (eq (6), at the bottom of page 2 of our main text) having a similar form as the objective function $\mathcal{L}_{S}$ in [NS]. A similar observation was also made by Yan et al., 2023, who optimize essentially the same functional. **We believe this equivalence holds for particular forms of the objective functional, but not more generally.** Particularly, in the case of WGD on the *EOT functional*, the objective $\mathcal{L}\_{EOT}$ (eq (7), after line 88 of our main text) is defined through an optimization and does not have a canonical formula (like the BA functional has in our eq (6)), so it is unclear how to even properly define the corresponding Euclidean descent scheme w.r.t. particle locations.
> >
> > 3. When WGD is equivalent to Euclidean GD on the particle locations (i.e., optimizing the BA functional and with uniformly-weighted particles), we agree it is possible to derive convergence of WGD by appealing to convergence results for the latter (Euclidean GD). However, when this equivalence fails to hold, such as performing WGD on the EOT functional, our Lemma 4.2 can still be applied to show convergence of Wasserstein gradient descent. Actually, given the right smoothness of the objective functional, the local convergence result of particle gradient descent is not unexpected. So in our opinion, the main contribution of our Lemma 4.2 is to identify and derive such regularity conditions for a given functional (namely eq (15) on line 202 of our text), which then enable a straightforward convergence argument. That argument (the four-line proof of Lemma 4.2) is indeed just one version of the usual arguments for gradient descent on smooth functions. Compared to one standard result — Theorem 3.2 of Nocedal and Wright 2006, which establishes the convergence of the sum of gradient norms — our version is slightly different in that the step sizes are multiplied with the gradient norms (see the last line of our proof), since the regularity conditions that we establish are slightly weaker than the usual ones.
> >
> >
> >
> > ----
> > **References**
> >
> > Atsushi Nitanda and Taiji Suzuki. Stochastic particle gradient descent for infinite ensembles. arXiv preprint arXiv:1712.05438, 2017
> >
> > Jorge Nocedal and Stephen J Wright. Numerical optimization. Springer, 2006
> >
> > Yan, Yuling, Kaizheng Wang, and Philippe Rigollet. Learning Gaussian mixtures using the Wasserstein-Fisher-Rao gradient flow. arXiv preprint arXiv:2301.01766, 2023.

---

### Decision · Program_Chairs · 2023-09-21

**Decision:**

Accept (poster)

**Comment:**

A novel method is proposed for computing the rate-distortion function with inspirations from Wasserstein gradient flows. This is an interesting bridge between optimal transport and information theory. Although the proposed method involves a non-convex optimization, experiments show the proposed method outperforms some existing methods. The authors addressed initial concerns in the rebuttal.